

# Lieb-Schultz-Mattis anomalies as obstructions to gauging (non-on-site) symmetries

Sahand Seifnashri⋆

School of Natural Sciences, Institute for Advanced Study,
1 Einstein Drive, Princeton, NJ 08540, USA

⋆ sahand@ias.edu

## Abstract

We study 't Hooft anomalies of global symmetries in 1+1d lattice Hamiltonian systems. We consider anomalies in internal and lattice translation symmetries. We derive a microscopic formula for the "anomaly cocycle" using topological defects implementing twisted boundary conditions. The anomaly takes value in the cohomology group $H^3(G, U(1)) \times H^2(G, U(1))$. The first factor captures the anomaly in the internal symmetry group $G$, and the second factor corresponds to a generalized Lieb-Schultz-Mattis anomaly involving $G$ and lattice translation. We present a systematic procedure to gauge internal symmetries (that may not act on-site) on the lattice. We show that the anomaly cocycle is the obstruction to gauging the internal symmetry while preserving the lattice translation symmetry. As an application, we construct anomaly-free chiral lattice gauge theories. We demonstrate a one-to-one correspondence between (locality-preserving) symmetry operators and topological defects, which is essential for the results we prove. We also discuss the generalization to fermionic theories. Finally, we construct non-invertible lattice translation symmetries by gauging internal symmetries with a Lieb-Schultz-Mattis anomaly.



# 1  Introduction

Global symmetry, and its generalizations, can impose powerful constraints on the dynamics of strongly coupled quantum systems, both in the continuum and on the lattice. The prime example in the continuum is the 't Hooft anomaly matching condition [2], and one of the earliest examples on the lattice is the Lieb-Schultz-Mattis (LSM) theorem [3]. Building upon recent progress, we establish a direct connection between such constraints on the lattice and in the continuum. Specifically, we consider the following three types of constraints and will phrase them in a unified framework.

**I. 't Hooft anomalies:** A 't Hooft anomaly of a global symmetry is defined as the obstruction to gauging the symmetry; see [4] for a recent review. The powerful constraint on the dynamics arises from the 't Hooft anomaly matching condition stating that the anomaly is invariant under the renormalization group (RG) flow [2]. In particular, it implies that a theory with an 't Hooft anomaly cannot flow to the trivial theory in the infrared (IR), which has no anomaly. Some of the early applications of anomaly matching appeared in the study of strongly coupled supersymmetric field theories [5]. Recently, 't Hooft anomalies of generalized symmetries [4,6–10] have constrained the dynamics of strongly coupled gauge theories in 3+1 [11] and 1+1 dimensions [12].

**II. The boundary of SPTs:** On the lattice, there exist similar constraints arising from "anomalies" of global symmetries. However, the definition of anomalies on the lattice differs from the continuum. The anomaly on the lattice is usually defined through the "bulk-boundary correspondence"[1] stating that anomalous theories correspond to boundaries of symmetry-protected topological (SPT) phases [17–19].[2]

---

[1]The continuum version of this correspondence is the anomaly inflow picture [13]. In modern terminology, it states that the 't Hooft anomaly of symmetry can be canceled by adding an *invertible field theory* [14] in the bulk. The combined bulk-boundary system is anomaly free and can be consistently coupled to background gauge fields. See [15,16] for reviews.

[2]There also exist anomalies on boundaries of "invertible" topological phases that are not protected by any symmetry. In the continuum, these anomalies correspond to gravitational 't Hooft anomalies. See [20] for a

A symmetry-protected topological (SPT) phase is a gapped phase with a symmetry where the non-triviality of the phase only depends on the symmetry [21–24]. More specifically, a non-trivial SPT has a unique gapped ground state that can be deformed into a trivial product state by a local Hamiltonian evolution that necessarily breaks the symmetry. Important examples of SPT phases are 2+1d and 3+1d topological insulators[3] [27–29], and odd-spin Haldane chain [30–32].

The lattice analog of 't Hooft anomaly matching condition is the fact that SPT phases have non-trivial symmetry-protected edge modes (see e.g. [33,34]). For instance, topological insulators have gapless edges protected by time-reversal and U(1) symmetries. Generally, the boundary of an SPT is either gapless, breaks the symmetry spontaneously, or has topological order.

**III. LSM anomalies:** A related powerful constraint on the lattice is the Lieb-Schultz-Mattis theorem [3,35] and its higher-dimensional generalizations [36,37]. The original LSM theorem states that translation-invariant 1+1d quantum spin chains with SO(3) symmetry, and half-integer spins per site, cannot have a unique gapped ground state; see [38] for a review. The LSM theorem has been further generalized to any translation-invariant system such that the internal symmetry acts projectively on each site [39–42]. For generalizations to fermionic systems see [43–46]; for other generalizations see [47,48].

Recently, it has been argued that the LSM-type theorems should be thought of as a consequence of a mixed anomaly between the internal symmetry and lattice translation [49–58]. For instance, the condition of having a projective representation of the internal symmetry per site was claimed to be an anomaly. We will refer to such conditions leading to generalized LSM theorems as LSM anomalies.

**The relation between I, II, and III**

There are two types of arguments in the literature relating LSM-type theorems with notions of anomalies described in **I** and **II** above. Related to the bulk-boundary correspondence, it has been argued that theories with LSM anomalies live on the boundary of *crystalline* SPT phases [49, 53, 59–62] protected by the translation and an internal symmetry. Therefore, the consequence of LSM-type theorems was phrased as the existence of anomalous edge modes of crystalline SPTs.

In a second line of reasoning, [50, 52, 57] argued that LSM anomalies lead to mixed 't Hooft anomalies between two *internal* symmetries after taking the continuum limit.[4] Note that lattice translation by one site usually becomes trivial in the continuum limit since the lattice spacing goes to zero in that limit. However, in some cases, such as when the result of the LSM theorem applies, lattice translation by one site becomes an internal symmetry in the continuum.

Building on [57], we provide a third perspective on the connection between the LSM theorem and 't Hooft anomalies. We prove that when there is an LSM anomaly, there is an obstruction to gauging the internal symmetry on the lattice while preserving the translation symmetry. This obstruction is the standard definition of 't Hooft anomaly.

In summary, recent developments have shown that various "lattice anomalies", defined in **II** and **III**, become 't Hooft anomalies after taking the continuum limit. However, so far there has not been a direct argument relating the definition of lattice anomalies to 't Hooft

---

discussion.

[3]In general, topological insulators/superconductors correspond to *non-interacting* fermionic SPT phases with/ without a U(1) symmetry, see [25, 26] for a complete classification.

[4]See also [63] for a discussion of the anomalies of the low-energy continuum theory in 2+1d.

anomalies. In this work, we show that even lattice anomalies are obstructions to gauging, thereby identifying lattice anomalies as 't Hooft anomalies.

We consider 1+1d lattice Hamiltonian systems and study anomalies in internal symmetries as well as LSM anomalies involving lattice translation. We introduce a procedure for gauging internal symmetries that may not act on-site on the lattice. Moreover, we present a simple formula to compute the anomaly locally and identify the anomaly as the obstruction to gauging. We demonstrate these results using topological defects that implement symmetry twists in space and establish a one-to-one correspondence between (locality-preserving) global symmetries and topological defects.

We mostly focus on bosonic systems and discuss the fermionic case in Section 4.4. Although we do not discuss it here, our method can capture anomalies involving other spatial symmetries, such as reflection and time-reversal symmetry.

This work is mainly inspired by the recent work of Cheng and Seiberg [57], where they compute lattice anomalies as anomalous phases arising from coupling the theory to background gauge fields. Here, we take one step further and quantify these anomalies. Moreover, our method for computing the anomaly is *local* and, therefore, insensitive to lattice details, such as the number of sites and boundary conditions.

We quantify the anomaly as a U(1)-valued function $F$, which is subject to some conditions and identifications. We refer to $F$ as the *anomaly cocycle* for its relation to group cohomology [64, 65]. We provide a systematic method to locally compute the anomaly cocycle on the lattice.[5] Our method does not assume prior knowledge of group cohomology; however, our result can be nicely phrased in this language, as we explain in the main text. Specifically, we identify the anomaly as an element of $H^3(G, U(1)) \times H^2(G, U(1)) \cong H^3(G \times \mathbb{Z}, U(1))$.

For the case of internal symmetries, our result is consistent with the proposed classification of 2+1d SPTs [21] and 1+1d 't Hooft anomalies in the continuum [19, 69].[6] For LSM anomalies, our result matches with the prediction from the conjectural "crystalline equivalence principle" of [59]. According to this conjecture, one can treat the lattice translation symmetry as an internal $\mathbb{Z}$ symmetry to classify possible anomalies.

Now we comment on the relation between our method and previous works on the computation of the anomaly cocycle on the lattice. For the case of internal symmetries in 1+1d, methods for computing the $H^3$ anomaly on the lattice were given in [1, 33, 78].[7] Chen, Liu, and Wen [33] provided a method to compute the anomaly when the symmetry is presented by a matrix product unitary operator [82]. Else and Nayak [1] computed the anomaly by considering truncated symmetry operators acting on subregions with boundaries. Their method applies to unitary symmetries that are representable as finite-depth quantum circuits (see, e.g., [83, 84] for the definition of finite-depth circuits). Finally, Kawagoe and Levin [78] introduced a method to compute the anomaly of arbitrary internal symmetries, including antiunitary symmetries. In their approach, they first choose a Hamiltonian that spontaneously breaks the symmetry, and then find the anomaly via the fusion of domain walls.

None of the methods mentioned above apply to LSM anomalies involving lattice translation. Our way of computing the anomaly is advantageous since it captures LSM anomalies. We find the LSM anomaly valued in $H^2(G, U(1))$ for arbitrary internal symmetry $G$, which does

---

[5]In general, there is no systematic method to compute arbitrary 't Hooft anomalies in the continuum. However, see [8, 66, 67] for the case of discrete symmetries of 1+1d CFTs, and see [68] for a method to detect certain global anomalies on the torus Hilbert space.

[6]More generally, anomalies are conjecturally classified by generalized cohomology theory on the lattice [70–72] (which is related to the decorated domain-wall construction [73]), and by cobordism groups in the continuum [74–77].

[7]A method to compute the $H^3$ anomaly was developed long ago in the context of algebraic quantum field theory [79–81]. Although stated in a different language, this method is the same as the one in [1]. We thank Yuji Tachikawa for pointing out these references to us.

not necessarily act on-site. Therefore, our results further generalize the LSM theorem beyond on-site symmetries [39–42]. Our results apply to locality-preserving symmetries, which are not necessarily finite-depth circuits.

Finally, let us point out that there are extensions of the original LSM theorem concerning translation-invariant systems with only a U(1) symmetry instead of the SO(3) symmetry. In particular, [85] found that certain translation-invariant spin chains cannot have a unique gapped ground state given that the ground state has a fractional U(1) charge per unit cell, see also [86]. More specifically, [85] studied the XXZ chain with a non-zero magnetic field. Such extensions are not associated with the standard notion of 't Hooft anomaly, that is, the obstruction to gauging. Instead, [57] phrase them as anomalies in the system with a fixed charge. Our formalism does not apply to such cases. This is mainly because U(1) does not have a genuine projective representation. Thus the notion of having a fractional U(1) charge per unit cell is an extra *constraint* (known as the filling constraint) rather than being a property of the microscopic system. See [87] for interesting results in systems with a continuously tunable filling.

We will summarize our results below, starting with a short overview of topological defects.

## 1.1 Topological defects

The 't Hooft anomaly is a microscopic property of the theory, independent of lattice details. To compute it locally, it is useful to consider symmetry *defects* that impose a symmetry action across space instead of symmetry *operators* that impose a symmetry action across time. Symmetries are traditionally defined by unitary operators that commute with the Hamiltonian. However, not every unitary operator that commutes with the Hamiltonian qualifies as a global symmetry. One further requires that the symmetry operator maps local operators to local operators. To manifest this locality property, it will be extremely useful to consider symmetry defects implementing symmetry twists in space.

Consider a 1+1d theory with a global symmetry. We can either put the system on a chain with periodic boundary conditions or impose twisted boundary conditions with respect to various elements of the symmetry group. For internal symmetries, twisted boundary conditions can be imposed *locally* by inserting symmetry defects. Defects generally correspond to the modification of the system locally near some point in space. Symmetry defects are, however, *topological* in the sense that we can move them locally without changing the physics. The crucial point is that we can capture all aspects of symmetries by the corresponding *topological defects*.[8]

For example, consider the transfers-field Ising model Hamiltonian

$$H = -\sum_j \sigma_j^z \sigma_{j+1}^z - h \sum_j \sigma_j^x. \tag{1}$$

The model has a global $\mathbb{Z}_2$ symmetry that flips the spins, denoted by $\eta$. The symmetry operator is given by $U_\eta = \prod_j \sigma_j^x$, which imposes a twist in time when inserted inside correlation functions. On the other hand, the symmetry defect inserted on link $(0, 1)$ is given by the defect Hamiltonian

$$H_\eta^{(0,1)} = -\sum_{j \neq 0} \left( \sigma_j^z \sigma_{j+1}^z \right) + \sigma_0^z \sigma_1^z - h \sum_j \sigma_j^x, \tag{2}$$

and implements a twist in space.

Topological defects capture all the properties of a global symmetry and its anomaly. For instance, the fact that $U_\eta$ generates a $\mathbb{Z}_2$ symmetry translates to a $\mathbb{Z}_2$ fusion rule for the defect,

---

[8]In a relativistic theory, topological defects extending in time are related to symmetry operators extending in space by Lorentz transformations. Even though there is no relativistic invariance in lattice Hamiltonian systems, topological defects are still related to symmetry operators, as we explain in Section 2.

i.e., $\eta \times \eta = 1$. This means that the insertion of two defects is trivial. Consider two defects at links $(0, 1)$ and link $(j, j+1)$; we can annihilate/create these defects by conjugating the Hamiltonian with the unitary operator $\sigma_1^x \sigma_2^x \ldots \sigma_j^x$.

In Section 2.4, we show that any internal symmetry $G$ can be represented by topological defects corresponding to a local modification of the Hamiltonian (and possibly the Hilbert space). We denote the defect Hamiltonian for the insertion of a symmetry defect for $g \in G$ on link $(j, j+1)$ by

$$H_g^{(j,j+1)} = \quad \text{---} \underset{j-1}{\bullet} \quad \underset{j}{\bullet} \; \overset{g}{\underset{j+1}{\times}} \; \underset{j+2}{\bullet} \text{---} . \tag{3}$$

The group multiplication of symmetry operators corresponds to the fusion of these defects. Consider two defects $g \in G$ and $h \in G$ on links $(j-1, j)$ and $(j, j+1)$, and denote the corresponding defect Hamiltonian by $H_{g;h}^{(j-1,j);(j,j+1)}$. These topological defects obey a $G$ fusion rule in the sense that there exist local unitary (fusion) operators

$$\lambda^j(g, h) = \qquad \qquad : H_{g;h}^{(j-1,j);(j,j+1)} \mapsto H_{gh}^{(j,j+1)}, \tag{4}$$

such that

$$H_{gh}^{(j,j+1)} = \lambda^j(g, h) H_{g;h}^{(j-1,j);(j,j+1)} \left( \lambda^j(g, h) \right)^{-1} . \tag{5}$$

## 1.2 Microscopic formula for the anomaly

As we will argue in Section 3, the 't Hooft anomaly is given by the relation

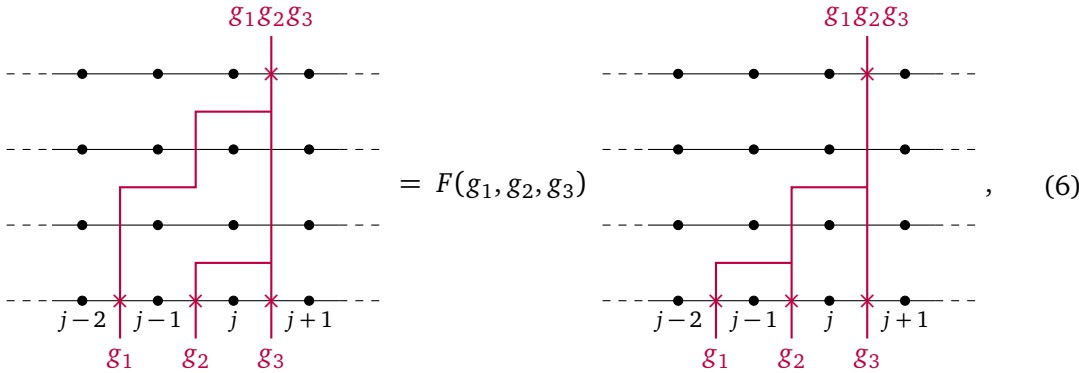

$$= F(g_1, g_2, g_3) \tag{6}$$

which in equation is

$$\lambda^j(g_1, g_2 g_3) \lambda^{j-1}(g_1, 1) \lambda^j(g_2, g_3) = F(g_1, g_2, g_3) \, \lambda^j(g_1 g_2, g_3) \lambda^{j-1}(g_1, g_2). \tag{7}$$

Here, $F(g_1, g_2, g_3)$ is just a phase and it is independent of $j$ due to the translation symmetry $T$ that satisfies $T \lambda^j(g, h) T^{-1} = \lambda^{j+1}(g, h)$. In Section 3.1, we discuss the general case that applies to systems without translation symmetry. We identify the anomaly as the function $F : G \times G \times G \to U(1)$ that is subject to the following two essential conditions:

**I. Cocycle condition:** $F$ satisfies a *modified* cocycle/pentagon equation

$$\frac{F(g_2,g_3,g_4)F(g_1,g_2g_3,g_4)F(g_1,g_2,g_3)}{F(g_1g_2,g_3,g_4)F(g_1,g_2,g_3g_4)} = F(g_1,g_2,1)\,. \tag{8}$$

This is the analog of the Wess-Zumino consistency conditions appearing in the classification of perturbative anomalies in the continuum.

**II. Phase ambiguity:** Fusion operators $\lambda^j(g,h)$, defined by equation (5), have phase ambiguity that propagate into $F$ and lead to the identification

$$F(g_1,g_2,g_3) \sim F(g_1,g_2,g_3)\frac{\gamma^j(g_2,g_3)\gamma^{j-1}(g_1,1)\gamma^j(g_1,g_2g_3)}{\gamma^{j-1}(g_1,g_2)\gamma^j(g_1g_2,g_3)}\,, \tag{9}$$

for $\gamma^j : G \times G \to U(1)$. These phase ambiguities correspond to local counterterms in the continuum that can be added into the action to cancel some anomalous phases.

Let us unpack these conditions into several comments:

1. The function $F$ defines the anomaly cocycles

$$\omega(g_1,g_2,g_3) = \frac{F(g_1,g_2,g_3)}{F(g_1,g_2,1)}\,, \qquad \text{and} \qquad \alpha(g_1,g_2) = F(g_1,g_2,1)\,, \tag{10}$$

satisfying the standard 3-cocycle and 2-cocycle conditions, respectively.

2. Modding out by $j$-independent phase ambiguities, which do not spoil the condition $T\,\lambda^j(g,h)\,T^{-1} = \lambda^{j+1}(g,h)$, we get the *cohomology classes* of $\omega$ and $\alpha$ denoted by

$$[\omega] \in H^3(G,U(1))\,, \qquad \text{and} \qquad [\alpha] \in H^2(G,U(1))\,. \tag{11}$$

If we further mod out by $j$-dependent phases, we completely trivialize $\alpha$ and are left with only $[\omega]$.

3. $[\omega] \in H^3(G,U(1))$ is the 't Hooft anomaly in $G$.

4. $[\alpha] \in H^2(G,U(1))$ is the (generalized) LSM anomaly, which is given by

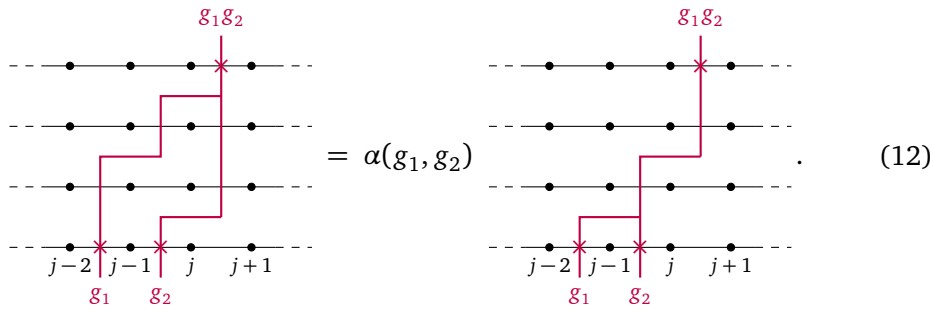

5. Without a translation symmetry $\omega$ and $\alpha$ might be $j$-dependent. However, the cohomology class $[\omega]$ is always $j$-independent and thus meaningful in systems without translation symmetry.

6. According to the Künneth formula $H^2(G,U(1)) \times H^3(G,U(1)) \cong H^3(\mathbb{Z} \times G, U(1))$. Therefore, we can identify $[F] = ([\alpha],[\omega])$ as an element of the third group cohomology of $\mathbb{Z} \times G$. This is consistent with the crystalline equivalence principle [59] stating that the translation symmetry can be treated as an internal $\mathbb{Z}$ symmetry to classify anomalies.

### 1.3 Anomaly as the obstruction to gauging

At least in the continuum, it is well-known that twisted boundary conditions for a symmetry $G$ correspond to coupling the theory to flat background $G$ gauge fields. The spatial component of the gauge field determines the twist in space or, equivalently, the insertion of a topological defect. Therefore, we can view the insertion of topological defects as coupling the system to background gauge fields and view the unitary fusion operators in equation (4) as gauge transformations.[9]

We will make the connection between topological defects and gauge fields even more concrete by explicitly gauging the symmetry on the lattice using topological defects. We develop a systematic way to gauge arbitrary internal symmetries without assuming that the symmetry is on-site or going to a presentation where the symmetry is on-site. This allows us to identify the anomaly cocycle as an obstruction to gauging even on the lattice.

More precisely, we will show that $[\omega] \in H^3(G, U(1))$ is the complete obstruction to gauging $G$ on the lattice. Therefore, it justifies calling $[\omega]$ the *'t Hooft* anomaly in $G$. This observation proves that the classification of 1+1d anomalies on the lattice and continuum are the same. In addition, we show that the (generalized) LSM anomaly $[\alpha] \in H^2(G, U(1))$ is the obstruction to gauging $G$ while preserving the lattice translation symmetry. Before ending this subsection, let us briefly sketch our gauging procedure and explain how the anomaly comes about as the obstruction.

To perform the gauging, we will first enlarge the Hilbert space by adding all possible $G$-defects on all links. This corresponds to making the gauge fields dynamical [6]. The next step is to impose Gauss's law locally at all sites. Roughly speaking, Gauss's law at site $j$ is given by the following projection operator

$$P_j \sim \frac{1}{|G|} \bigoplus_{gh=g'h'} \quad$$ 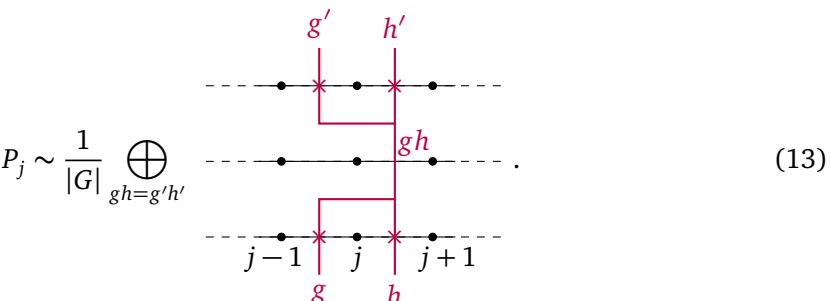 $$. \tag{13}$$

See equations (101) and (104) for details.

However, we can consistently impose Gauss's law constraints if they commute with each other. In other words, if $P_j$ commutes with $P_{j+1}$. As we show in Section 3.2, this holds if and only if the anomaly cocycles $\omega$ and $\alpha$ vanish. However, since the anomaly cocycles have phase ambiguities, the true obstruction is given by their cohomology classes $[\omega]$ and $[\alpha]$.

Without a translation symmetry, we can redefine the fusion operators by $j$-dependent phases to eliminate the 2-cocycle $\alpha$. Thus we establish that $[\omega]$ is the complete obstruction to gauging $G$.

With a translation symmetry, however, we cannot redefine the fusion operators with $j$-dependent phases, as that would violate the relation $TP_jT^{-1} = P_{j+1}$. Breaking such a relation means that $T$ is not gauge invariant, and we lose the translation symmetry by one site. Therefore, if we insist on preserving the translation symmetry, the obstruction to gauging $G$ is given by both $[\omega]$ and $[\alpha]$. Therefore, we identify the LSM anomaly $[\alpha]$ as the mixed 't Hooft anomaly between $G$ and the translation symmetry.

---

[9]This point was emphasized in [6], and [57] used this approach to detect anomalies.

**Anomaly matching on the lattice**

The 't Hooft anomaly matching states that the anomaly of the Ultraviolet (UV) physics should match with the anomaly of the Infrared (IR) low-energy physics. In the continuum, this follows from the fact that the 't Hooft anomaly is invariant under continuous deformations, and in particular, under the renormalization group (RG) flow. Therefore, a theory with an 't Hooft anomaly in the UV cannot flow to a trivial theory in the IR since the trivial theory has no anomaly.

There are usually three options for the low-energy physics when there is a non-trivial 't Hooft anomaly. Either we have a gapless theory, ground state degeneracy, or topological order. However, there is no topological order in 1+1d. We will prove an 't Hooft anomaly matching on the lattice by showing the following. We will show that a 1+1d lattice Hamiltonian model with a non-trivial 't Hooft anomaly, i.e., $[\omega] \neq 1$ or $[\alpha] \neq 1$, cannot have a unique gapped ground state. This statement generalizes the LSM theorem and can be viewed as a lattice version of the 't Hooft anomaly matching conditions.

**Anomaly matching in the continuum**

Now we discuss the 't Hooft anomaly matching between the UV lattice model and the IR theory in the continuum. As usual, there is a homomorphism from the symmetry of the UV into the symmetry group in the IR. However, the generator of the translation symmetry, $T$, either becomes trivial in the continuum or becomes an internal symmetry of the IR. See [57] for a detailed discussion and examples. Therefore to discuss this homomorphism, we only need to consider the internal symmetries of the IR.

If we denote the *internal* global symmetry of the IR theory by $G_{\text{IR}}$, we get a homomorphism

$$f : \mathbb{Z} \times G \to G_{\text{IR}}. \tag{14}$$

This homomorphism captures the continuum limit of translation by one site, which is an internal symmetry in the continuum. Specifically, we consider translation by one site and then take the continuum limit. If, on the other hand, we consider translations by $n$ sites and take the number of lattice sites $L$ to infinity such that $n/L$ is constant, we find translations of the continuum theory.[10]

We claim that the UV anomaly is given by the *pullback* of the IR anomaly via the homomorphism $f$. If we denote the 't Hooft anomaly of the IR theory by $[\omega_{\text{IR}}] \in H^3(G_{\text{IR}}, U(1))$, in equations, we get

$$f^*([\omega_{\text{IR}}]) = ([\alpha], [\omega]) \in H^3(\mathbb{Z} \times G, U(1)). \tag{15}$$

Equation (15) is a precise mathematical statement of the anomaly matching. In particular, it implies that when the UV anomaly is non-trivial, the IR anomaly $[\omega_{\text{IR}}]$ must also be non-trivial. Moreover, as emphasized in [57], the LSM anomaly $[\alpha]$ becomes a mixed 't Hooft anomaly between the symmetry $G$ and the *emanant* symmetry of the IR. The emanant symmetry is the symmetry generated by the continuum limit of translation by one site.

## 1.4 Outline

In Section 2, we discuss the connection between symmetry operators and topological defects. In particular, we prove a one-to-one correspondence between locality-preserving global symmetries and topological defects in 1+1d lattice Hamiltonian systems. This correspondence will be essential for the results we prove in Section 3.

---

[10]We thank Nati Seiberg for a discussion on this point.

Section 3.1 presents our microscopic formula for 't Hooft anomalies and discusses its relation to the group cohomology classification of anomalies. Moving on to Section 3.2, we introduce a method to gauge non-on-site internal symmetries on the lattice. We identify the anomaly as the obstruction to this gauging procedure. In Section 3.3, we discuss a generalized LSM-type constraint stating that when the anomaly is non-trivial, there is no symmetric deformation to a gapped phase with a unique ground state.

Section 4 contains several examples illustrating our general results. In Section 4.1, we discuss LSM anomalies for on-site symmetries, particularly the spin-half Heisenberg chain example. In Section 4.2, we study the XYZ chain and its $\mathbb{Z}_2 \times \mathbb{Z}_2$ symmetry, which has an LSM anomaly. We construct a non-invertible lattice translation symmetry by gauging the $\mathbb{Z}_2 \times \mathbb{Z}_2$ symmetry of the XYZ model. In Section 4.3, we consider the modified Villain theory that exhibits anomalous U(1) chiral symmetries on the lattice. We use this theory to construct Abelian chiral gauge theories on the lattice. In Section 4.4, we discuss the fermionic generalization of the results from Section 3 and demonstrate it by considering the example of a chain of $N$ free Majorana fermions.

Finally, Appendix A provides a single formula to extract the 't Hooft anomaly in internal $\mathbb{Z}_2$ symmetries. In Appendix B, we compare our results with Else-Nayak [1]. We show that our formula of the $H^3$ anomaly matches Else-Nayak in cases where their method is applicable. In Appendix C, we construct topological defects for antiunitary symmetries.

## 2 Symmetry operators vs. topological defects

In this section, we discuss the precise relation between topological defects and global symmetries of 1+1d lattice Hamiltonians. In particular, we construct topological defects from internal symmetry operators and, conversely, construct symmetry operators from topological defects. More generally, we show a one-to-one correspondence between topological defects and *locality-preserving* symmetry operators, which include lattice translation and antiunitary symmetries.

### 2.1 Symmetry operators

Symmetries of quantum systems are traditionally defined by unitary operators that act on the Hilbert space and commute with the Hamiltonian. This is a complete characterization of symmetries in quantum mechanics. In lattice models with spatial dimensions, there is a notion of locality and one requires symmetries to act locally. More precisely, the symmetry operator must map local operators into local operators.[11]

The simplest way to satisfy the locality requirement is to consider symmetries that act on-site. Consider a system with a tensor product Hilbert space $\mathcal{H} = \bigotimes_j \mathcal{H}_j$ where $\mathcal{H}_j$ is the Hilbert space at site $j$. An on-site symmetry operator takes the form

$$U = \prod_j U_j, \tag{16}$$

where $U_j$ is a unitary operator that acts non-trivially only on site $j$. Such an operator respects locality since it maps a local operator $O_j$ supported on site $j$ to $U O_j U^{-1} = U_j O_j U_j^{-1}$, which is also supported on site $j$. The operators $O_j$ and $U_j O_j U_j^{-1}$ are considered local operators since they are supported on a single site.

---

[11]In the continuum such an operator is called a line operator. Its locality is manifest if it is defined by a local action $U(\gamma) = e^{-i \oint_\gamma \mathcal{L}[\phi]}$, such as Wilson lines in gauge theories.

More generally, we require symmetries to act locally in the following sense. For a symmetry operator $U$, there should exist an integer $k$ – independent of the number of lattice sites – such that $U$ maps operators supported on $l$ consecutive sites to operators that have support on at most $l + k$ consecutive sites.[12] According to this definition reflection across site $J$ acts locally, with $k = 0$, since it maps an operator supported on site $j$ to an operator supported on site $J - j$. However, the reflection symmetry is not *locality preserving*. A unitary operator is said to be locality preserving if it maps a local operator around site $j$ to a local operator which also has support around site $j$.[13]

In this work, we consider finite-range/local Hamiltonian systems in the sense that the Hamiltonian is written as a sum of local terms. Specifically, we assume the Hamiltonian is written as

$$H = \sum_j H_j \,, \tag{17}$$

where $H_j$ is a local operator with support on at most sites $j, j+1, \cdots, j+l$, for some integer $l$ independent of the system size. We assume the Hilbert space is constructed as a tensor product of local factors plus additional local constraints. For instance, there are local Gauss's law constraints in gauge theories. The list of local operators in such theories is given by operators that commute with local constraints. The locality is encoded in the algebra of local operators instead of the Hilbert space. More precisely, there exists a well-defined algebra of local operators such that each operator has a support consisting of a finite number of lattice sites. Operators with disjoint support necessarily commute with each other.[14]

It is the locality requirement that makes global symmetries powerful. The goal of this section is to give an alternative definition of global symmetries that makes the locality property manifest. This alternative definition is the notion of topological defects.

## 2.2 Topological defects

Symmetry operators when inserted inside correlation functions impose a symmetry action across a point in time, whereas symmetry *defects* impose a symmetry twist across space. Symmetry operators/defects are said to be *topological* in the sense that the correlation functions are independent of their exact location [6, 88]. Let us explain this terminology in the continuum setting, where it is more natural.

In the continuum, one can insert symmetry operators/defects on arbitrary loops, i.e. codimension-one submanifolds, inside the 1+1d spacetime. The correlation functions only depend on the topology of the loop and are insensitive to local deformations of the loop. More precisely, consider the symmetry operator $U(\gamma)$ supported on the loop $\gamma$. One can deform the loop $\gamma$ slightly and the correlation functions do not change unless we cross a local operator insertion $O(x)$. In that case, swiping $\gamma$ across the local operator insertion at $x$ implements the symmetry action on that local operator. Therefore, we have the following relation characterizing the symmetry through the correlation functions of topological operators/defects:

$$\left\langle \cdots \quad \overset{O(x)}{\underset{U(\gamma)}{\bigg\uparrow}} \quad \cdots \right\rangle = \left\langle \cdots \quad \overset{O'(x)}{\underset{U(\gamma')}{\phantom{x}}} \quad \cdots \right\rangle. \tag{18}$$

---

[12]In the case of a tensor product Hilbert space it is enough to check this condition only for $l = 1$. In other cases like gauge theories with Gauss's constraints, the gauge invariant operators supported on a single site do not necessarily generate all local operators.

[13]Locality-preserving unitary operators acting on lattices are called quantum cellular automata (QCAs).

[14]For fermionic theories, there exists a $\mathbb{Z}_2$ automorphism of the algebra of local operators given by the fermion parity. In that case, fermionic operators with disjoint support anti-commute each other.

Here, deforming the loop $\gamma$ into $\gamma'$ implements the symmetry action: $O(x) \mapsto O'(x)$.

In a lattice Hamiltonian system, there are two important cases. When $\gamma$ is located at a fixed time and extends in space, $U(\gamma)$ is the unitary operator that commutes with the Hamiltonian. However, when $\gamma$ is located at a point in space and extends in time, it defines a topological defect. As we will show later, topological defects are still related to symmetry operators even in non-relativistic lattice Hamiltonian systems.

Since symmetry defects impose symmetry twists across the space, their insertion corresponds to twisted boundary conditions on a closed lattice. Consider a 1+1d system with an internal $\mathbb{Z}_2$ symmetry. We can put the system on a closed chain with a periodic boundary condition or anti-periodic boundary condition for the $\mathbb{Z}_2$ symmetry. Imposing an anti-periodic boundary condition usually corresponds to changing the sign of a single term in the Hamiltonian. This local modification of the Hamiltonian near a single site is an example of a topological defect.

For example, consider the transfers-field Ising model (TFIM) Hamiltonian

$$H = -\sum_j \sigma_j^z \sigma_{j+1}^z - h \sum_j \sigma_j^x. \tag{19}$$

This model has a $\mathbb{Z}_2$ spin-flip symmetry that we denote by $\eta$. It is generated by the unitary operator $U_\eta = \prod_j \sigma_j^x$ that flips all the spins, i.e. $U_\eta$ conjugates $\sigma_j^z$ to $-\sigma_j^z$ for all $j$. We can introduce a $\mathbb{Z}_2$ twist at link $(0,1)$ by acting with the symmetry only on sites $j \geq 1$. By doing such, we find the twisted/defect Hamiltonian

$$H_\eta^{(0,1)} = -\sum_{j \neq 0} \sigma_j^z \sigma_{j+1}^z + \sigma_0^z \sigma_1^z - h \sum_j \sigma_j^x. \tag{20}$$

By identifying $\sigma_L = \sigma_0$, we find the model with an anti-periodic boundary condition on the chain with $L$ sites.

Note that by doing a change of basis (i.e. a similarity transformation), we can move the location of the defect from link $(0,1)$ to any other link of the chain. For instance, by conjugating the Hamiltonian with $\sigma_1^x$ or $\sigma_0^x$ we can move the defect, respectively, to the right or left

$$H_\eta^{(1,2)} = \sigma_1^x H_\eta^{(0,1)} \sigma_1^x, \qquad H_\eta^{(-1,0)} = \sigma_0^x H_\eta^{(0,1)} \sigma_0^x. \tag{21}$$

Therefore, the defect can be moved *locally* (i.e. with a local unitary operator) without costing any energy. Because of this property, we say that $\eta$ is a *topological* defect.[15] Mathematically speaking, the twisted boundary condition corresponds to the non-trivial $\mathbb{Z}_2$ bundle over the circle, and moving the topological defect is a gauge transformation.

The important property of topological defects is that we can use them to reconstruct the symmetry operators. To do such, let us begin with the untwisted Hamiltonian $H$ on a periodic chain. Conjugating $H$ with $\sigma_1^x$ creates a pair of defects on links $(L,1)$ and $(1,2)$

$$H_{\eta;\eta}^{(L,1);(1,2)} \equiv \sigma_1^x H \sigma_1^x = -\sum_{j \neq L,1} \sigma_j^z \sigma_{j+1}^z + \sigma_L^z \sigma_1^z + \sigma_1^z \sigma_2^z - h \sum_j \sigma_j^x. \tag{22}$$

Now we can apply the movement operators $\sigma_2^x, \sigma_3^x, \ldots, \sigma_{L-1}^x$ respectively to move the defect on link $(1,2)$ along the chain to link $(L-1,L)$. Finally, we apply $\sigma_L^x$ to annihilate the two defects on links $(L-1,L)$ and $(L,1)$, and return to the untwisted Hamiltonian. Putting everything together, we find that the product of all these unitaries conjugates $H$ back to itself and thus is a symmetry operator. Indeed this unitary operator is the original $\mathbb{Z}_2$ symmetry operator $U_\eta$.

---

[15]A classical example of this is the Mobius band. We can take a tape and cut it at some point and glue it back to itself with a $\pi$ rotation. The resulting Mobius band is translationally invariant; therefore, we cannot detect the location of the cut/defect.

The Ising chain and its $\mathbb{Z}_2$ symmetry explain the basic idea behind the relation between topological defects and symmetry operator. We will now define the notion of topological defects more abstractly and later show the relation between topological defects and symmetry operators in general.

**General definition of topological defects**

Generally, a defect corresponds to a modification of the theory only near some region. Physically, it corresponds to modifying a material locally by adding impurities, removing atoms, etc. In this work, we are interested in defects that are localized near a point in space.[16] We model such defects by modifying the Hamiltonian and the Hilbert space only in a local neighborhood of some sites. We refer to such a modified Hamiltonian and Hilbert space as the *defect Hamiltonian* and the *defect Hilbert space*.

Consider a general 1+1d lattice Hamiltonian system. We represent a defect by a list of defect Hamiltonians and defect Hilbert spaces representing the insertion of the defect on various links of the chain. For the system with a defect $\mathcal{D}$ on link $(j, j+1)$, we denote the defect Hilbert space and defect Hamiltonian as $\mathcal{H}_{\mathcal{D}}^{(j,j+1)}$ and $H_{\mathcal{D}}^{(j,j+1)}$, and represent it pictorially by

$$H_{\mathcal{D}}^{(j,j+1)} = \;\; \begin{array}{c} \mathcal{D} \\ \text{- - -} \bullet \!\!\!\!\!\!\!\!\!\!\!\!\! \underset{j-1}{} \!\!\!\!\!\!\!\!\!\!\bullet \!\!\!\!\!\!\!\!\!\underset{j}{} \!\!\!\!\!\!\! \times \!\!\!\!\!\!\bullet \!\!\!\!\!\!\!\!\!\!\underset{j+1}{} \!\!\!\!\!\!\!\!\!\!\bullet \!\!\!\!\!\!\!\!\!\!\underset{j+2}{} \text{- - -} \end{array} \; . \tag{23}$$

The defect $\mathcal{D}$ is local in the sense that $\mathcal{H}_{\mathcal{D}}^{(j,j+1)}$ and $H_{\mathcal{D}}^{(j,j+1)}$ only differ by the original Hilbert space $\mathcal{H}$ and Hamiltonian $H$ near link $(j, j+1)$.[17]

We say that a defect is topological if moving it does not change the physics. More precisely, the defect $\mathcal{D}$ is topological if there exist local unitary operators $U_{\mathcal{D}}^j : \mathcal{H}_{\mathcal{D}}^{(j-1,j)} \to \mathcal{H}_{\mathcal{D}}^{(j,j+1)}$ such that

$$H_{\mathcal{D}}^{(j,j+1)} = U_{\mathcal{D}}^j H_{\mathcal{D}}^{(j-1,j)} (U_{\mathcal{D}}^j)^{-1} . \tag{24}$$

The movement operator $U_{\mathcal{D}}^j$ is local in the sense that it has support on at most $k+1$ sites near site $j$. More precisely, we assume $U_{\mathcal{D}}^J$ has support on sites $J - k/2 \le j \le J + (k+1)/2$. Moreover, $k \ge 0$ is independent of the system size, and the smallest such integer is denoted as the *width* of the defect. In the Ising example, we have $U_{\eta}^j = \sigma_j^x$, $\mathcal{H}_{\eta}^{(j,j+1)} = \mathcal{H}$, and $k = 0$.

Importantly, the presentation of the defect in terms of defect Hamiltonians and defect Hilbert spaces is not unique. One can conjugate the defect Hamiltonians by local unitary operators $O_{\mathcal{D}}^{(j,j+1)}$ to find another presentation of the defect

$$H_{\mathcal{D}'}^{(j,j+1)} = O_{\mathcal{D}}^{(j,j+1)} H_{\mathcal{D}}^{(j,j+1)} (O_{\mathcal{D}}^{(j,j+1)})^{-1} . \tag{25}$$

The support of the local operator $O_{\mathcal{D}}^{(j,j+1)}$ determines the width of the defect in the new presentation. The movement operators in the new presentation are given by

$$U_{\mathcal{D}'}^j = O_{\mathcal{D}}^{(j,j+1)} U_{\mathcal{D}}^j (O_{\mathcal{D}}^{(j-1,j)})^{-1} . \tag{26}$$

In other words, we say that two defects $\mathcal{D}$ and $\mathcal{D}'$ are equivalent if the corresponding defect Hamiltonians are related by local unitary operators. In practice, we *fix* a particular presentation of the defects and keep in mind that physically meaningful quantities should be independent of this choice.

---

[16]From the space point of view they may be called point defects, but correspond to line defects in spacetime.

[17]More precisely, the algebra of local operators supported on sites $J \le j - k/2$ and $J \ge j + k/2$ of the system with and without the defect are isomorphic for a fixed integer $k$ related to the width of the defect defined below. This isomorphism identifies the local terms in the Hamiltonians $H_{\mathcal{D}}^{(j,j+1)}$ with those in $H$ on sites $J \le j - k/2$ and $J \ge j + k/2$.

**Fusion of topological defects**

By applying a sequence of movement operators, one can move topological defects arbitrarily around the chain. In particular, consider a system with two topological defects that are far away from each other. Using the movement operator, we can bring one of the defects next to the other one, thus defining a fusion between topological defects. In particular, for a symmetry $G$, inserting two symmetry twists associated with group elements $g \in G$ and $h \in G$ is equivalent to a single $gh \in G$ twist. In general, the fusion of symmetry defects corresponds to the group structure of symmetry transformations.

However, the fusion operation between arbitrary topological defects is not always group-like. In particular, while the fusion operation is associative, it is not necessarily invertible. In this work, we will only consider invertible topological defects with a group-like fusion rule. But let us point out that there exist non-invertible topological defects even in the transverse-field Ising model [89, 90].[18] For general construction of 1+1d lattice systems with non-invertible topological defects see [95–98].

To describe the fusion operation concretely, we want to consider a system with two defects located at nearby links. However, we do not want to introduce a new ambiguity in defining the defect Hamiltonian for two defects. In other words, starting from single-defect Hamiltonians, we want to unambiguously construct the defect Hamiltonian for the system with two defects. We will now argue that this is possible if the distance between the defects is larger than the width of each defect.

Consider two topological defects $g$ and $h$ of width at most $k$. The key property is that $U_g^J$ commutes with $U_h^{J+k+1}$ in the sense that

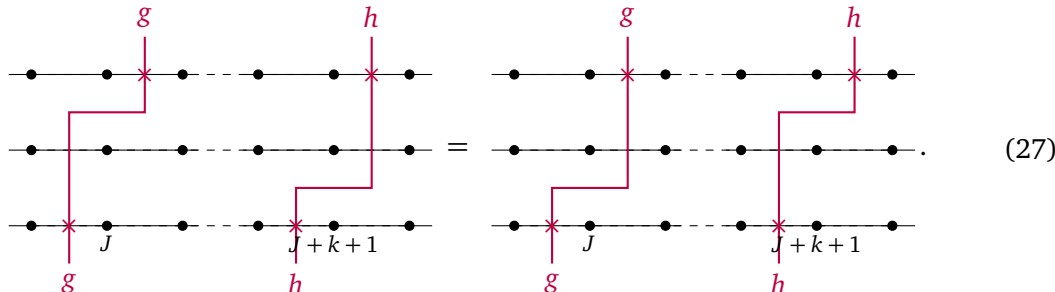

$$\tag{27}$$

In this case, it is unambiguous to insert the defects at links $(j, j+1)$ and $(j+k, j+k+1)$. We first insert them very far away from each other, say at links $(-J, -J+1)$ and $(J, J+1)$ for $J \gg |j| + k$. This is unambiguous since the defects are localized. We denote the corresponding defect Hamiltonian as $H_{g;h}^{(-J,-J+1);(J,J+1)}$. Now we apply the movement operators

$$U_g^j \cdots U_g^{-J+2} U_g^{-J+1}, \quad \text{and} \quad (U_h^{j+k+1})^{-1} \cdots (U_h^{J-1})^{-1} (U_h^J)^{-1}, \tag{28}$$

to bring them on links $(j, j+1)$ and $(j+k, j+k+1)$. There can be an ambiguity in doing such if the two movement operators do not commute with each other. However, since the defects have width $k$ the two movement operators commute with each other and thereby there is no ambiguity in defining $H_{g;h}^{(j,j+1);(j+k,j+k+1)}$.

From now on we assume that all the defects have width $k = 1$ such that we can always insert them, unambiguously, on adjacent links. However, everything that follows can be generalized to the case of $k > 1$. We consider general invertible topological defects whose fusion rule is governed by a group $G$ and label the defects by the group elements $g, h, k, \cdots \in G$. We

---

[18]Topological defects with a non-invertible fusion rule first appeared in the context of 1+1d rational conformal field theories [91–93], and later interpreted as generalized symmetries in [7, 8]; see [4, 10, 94] for reviews.

construct the defect Hamiltonian for the system with two defects $g$ and $h$ on links $(j-1,j)$ and $(j,j+1)$ and denote it as

$$H_{g;h}^{(j-1,j);(j,j+1)} = \quad \text{———} \underset{j-1}{\bullet} \overset{g}{\underset{j}{\times}} \bullet \overset{h}{\underset{j+1}{\times}} \bullet \text{———} . \tag{29}$$

**Fusion operators:** The fusion is implemented by unitary operators $\lambda^j(g,h): \mathcal{H}_{g;h}^{(j-1,j);(j,j+1)} \to \mathcal{H}_{gh}^{(j,j+1)}$, which are represented by the following spacetime diagram[19]

$$\lambda^j(g,h) = \qquad \qquad \qquad \qquad : H_{g;h}^{(j-1,j);(j,j+1)} \mapsto H_{gh}^{(j,j+1)} . \tag{30}$$

The local unitaries $\lambda^j(g,h)$ are called *fusion operators* and they satisfy

$$H_{gh}^{(j,j+1)} = \lambda^j(g,h) H_{g;h}^{(j-1,j);(j,j+1)} \left(\lambda^j(g,h)\right)^{-1} . \tag{31}$$

The movement operators of equation (24), are special cases of the fusion operators in the sense that $U_g^j = \lambda^j(g,1)$. Here, the identity element $1 \in G$ corresponds to the trivial defect.

**Simple defects:** The fusion operators satisfying (31) are well-defined up to phases. This is because we assume there is no non-trivial local unitary operator that commutes with defect Hamiltonians. If this had not been the case, we could have diagonalized the defect Hamiltonian with respect to such local unitaries and thereby decompose the defect into simpler ones. In such cases, the defect is said to be a non-simple, or reducible, defect. We will only consider *simple* topological defects in this work. Since the defects are invertible, the assumption that the defects are simple is equivalent to assuming that no local operator commutes with the untwisted Hamiltonian, i.e. the trivial defect is simple.[20]

Changing the presentation of the defects, given by (25), transforms the fusion operator

$$\lambda^j(g,h), \qquad \text{into} \qquad O_{gh}^{(j,j+1)} \lambda^j(g,h) \left(O_g^{(j-1,j)} O_h^{(j,j+1)}\right)^{-1} . \tag{32}$$

It is important to work with a fixed presentation of the defect Hamiltonians. At the end of the day, the only physically meaningful quantities are those independent of the defect's presentation and the phase ambiguities mentioned above.

Let us summarize the key properties of invertible topological defects (of width $k=1$):

1) There is a list of defect Hamiltonians $H_g^{(j,j+1)}$ for every $g \in G$ and link $(j,j+1)$.

2) There exist fusion operators $\lambda^j(g,h): H_{g;h}^{(j-1,j);(j,j+1)} \mapsto H_{gh}^{(j,j+1)}$.

3) $\lambda^j(g,h)$ are well-defined up to a phase.

4) $\lambda^j(g_1,g_2)$ commutes with $\lambda^{j+2}(g_3,g_4)$.

---

[19]We always use the notation '$\mapsto$' to denote conjugation, i.e. $U : O \mapsto UOU^{-1}$.

[20]We only consider simple defects since the classification of anomalies is different for non-simple defects. In the continuum, this corresponds to the fact that non-simple defects lead to 1-form symmetries, and anomalies of 0-form symmetries depend on 1-form symmetries. See [99, 100] for related discussions on 1-form symmetries.

**Internal symmetries vs. spatial symmetries**

So far, we have described the most general notion of invertible topological defects whose fusion is governed by a group $G$. To ensure that these topological defects correspond to internal symmetries, as opposed to spatial symmetries such as translation and reflection, we need to consider a further constraint on these defects. Let us first recall the definition of internal symmetries.

Global symmetries map local operators to local operators. If the symmetry is internal, it should map local operators supported around site $j$ to local operators that are also supported near site $j$. This criterion, however, is not sharp enough to distinguish between approximately internal symmetries, such as translation by one site, and true internal symmetries. A sharper definition is as follows.

We say that a symmetry $U$ is internal if for all integers $n$, $U^n$ maps a local operator near site $j$ to a local operator near site $j$. More precisely, $U$ generates an internal symmetry if there exists a non-negative integer $k$ independent of $n$ such that $U^n$ maps an operator supported on sites $J_1 < j < J_2$ to an operator that has support on at most sites $J_1 - k < j < J_2 + k$. Here, $k$ is independent of the system size, and we refer to the smallest such integer as the *width* of the symmetry.[21] The case of $k = 0$, corresponds to an on-site symmetry (which may or may not act projectively on local Hilbert spaces at each site).

Below, we will argue that a topological defect of width $k$ corresponds to a symmetry operator of width at most $2k$ and vice versa. Therefore, a topological defect $\mathcal{D}$ corresponds to an internal symmetry if there is an integer $k$ such that the width of $\mathcal{D}^n$ is at most $k$ for all $n \in \mathbb{Z}$.

## 2.3 Defects → operators

Here we will construct symmetry operators from invertible topological defects.

We start with the untwisted Hamiltonian $H$ for the periodic chain with $L$ sites. We first conjugate the untwisted Hamiltonian with $\left(\lambda^1(g^{-1}, g)\right)^{-1}$ to create a pair of $g^{-1}$ and $g$ defects on links $(L, 1)$ and $(1, 2)$. Then we apply the movement operator $\lambda^{L-1}(g, 1) \cdots \lambda^3(g, 1) \lambda^2(g, 1)$ to bring the defect $g$ around the chain to the other side of the defect $g^{-1}$. Finally, we fuse the two defects to get back the untwisted Hamiltonian. In summary, we find the line operator

$$U_g = \lambda^L(g, g^{-1}) \lambda^{L-1}(g, 1) \cdots \lambda^3(g, 1) \lambda^2(g, 1) \left(\lambda^1(g^{-1}, g)\right)^{-1}, \tag{33}$$

which commutes with the untwisted Hamiltonian $H$. Diagrammatically, we have

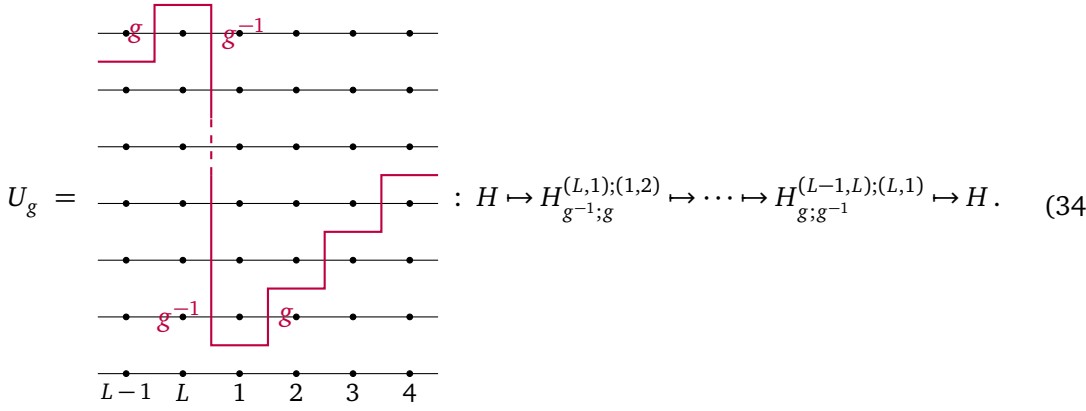

$$U_g = \qquad : H \mapsto H_{g^{-1};g}^{(L,1);(1,2)} \mapsto \cdots \mapsto H_{g;g^{-1}}^{(L-1,L);(L,1)} \mapsto H. \tag{34}$$

We now argue that the symmetry operator $U_g$ constructed above is unique up to a phase, and does not depend on various choices that were made in defining it. First of all, changing

---

[21]The width of a unitary operator defined here is closely related to the 'range' of that operator defined in the quantum cellular automata literature, see e.g. [84].

the presentation of the defect, as described in equation (32), does not affect $U_g$. Secondly, we show that $U_g$ does not depend on the choice of origin. In other words, we want to show that $U_g$, up to a phase, is equal to

$$\lambda^1(g, g^{-1})\lambda^L(g, 1)\cdots\lambda^4(g, 1)\lambda^3(g, 1)\left(\lambda^2(g^{-1}, g)\right)^{-1}. \tag{35}$$

We first note that we have the following relation

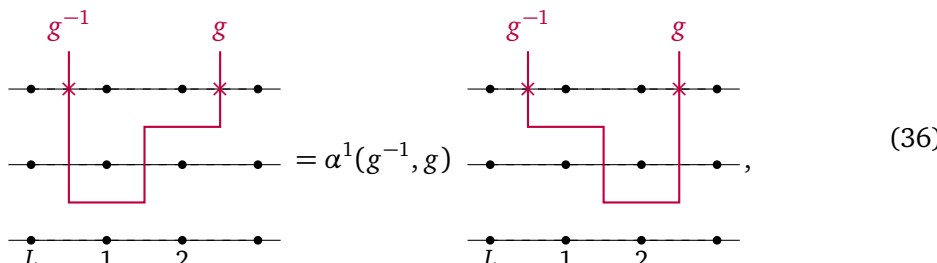

$$\tag{36}$$

where

$$\alpha^1(g^{-1}, g) = \lambda^2(g^{-1}, g)\lambda^1(g^{-1}, 1)\lambda^2(g, 1)\left(\lambda^1(g^{-1}, g)\right)^{-1}. \tag{37}$$

Here, $\alpha^1(g^{-1}, g)$ is a local unitary operator that commutes with the Hamiltonian, therefore by our assumption, it should be just a $U(1)$ phase.

Using the relation above, we can commute $\lambda^1$ pass through $\lambda^2$ in the right-hand side of equation (33). Moreover, assuming that the defect has width $k = 1$, we can commute $\lambda^1$ pass through other terms and bring it to the right of $\lambda^L$. Finally, we define

$$\alpha^L(g, g^{-1}) = \lambda^1(g, g^{-1})\lambda^L(g, 1)\lambda^1(g^{-1}, 1)\left(\lambda^L(g, g^{-1})\right)^{-1}, \tag{38}$$

which is also a phase similar to $\alpha^1$. This allows us to commute $\lambda^1$ through $\lambda^L$ to get

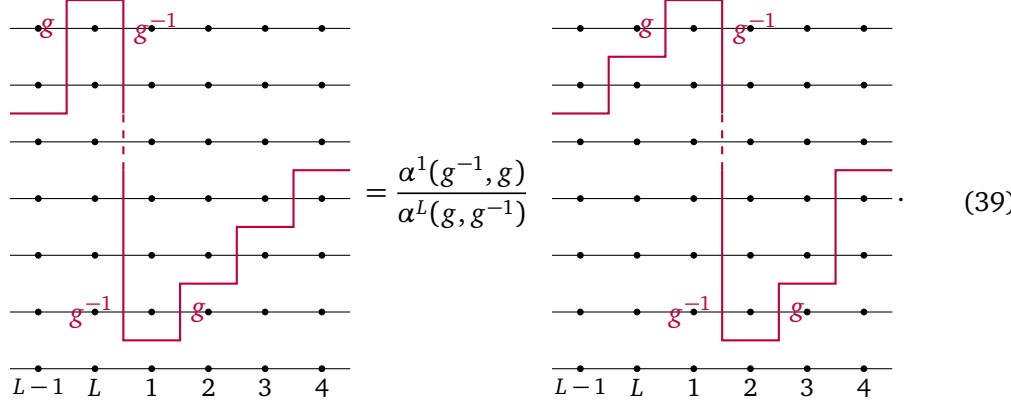

$$= \frac{\alpha^1(g^{-1}, g)}{\alpha^L(g, g^{-1})} \tag{39}$$

Therefore, we see that changing the origin in the definition of $U_g$ multiplies it by a phase. Thus the topological defect determines $U_g$ up to a phase. Now we show $U_g$ has the same width as the defect.

Assume that the defect $g$ has width $k$, so that $\lambda^J(g, h)$ has support on, at most, sites $J - k/2 \leq j \leq J + (k+1)/2$. We show that in this case the width of the unitary symmetry operator $U_g$, constructed above, is at most $k$. Recall that a symmetry operator has width $k$, if it maps an operator $O_{[J_1, J_2]}$ that has support on sites $J_1 \leq j \leq J_2$ into an operator that has support on, at most, sites $J_1 - k \leq j \leq J_2 + k$.

We notice that most of the terms in (33), besides those that are near sites $J_1$ and $J_2$, commute with $O_{[J_1, J_2]}$. Therefore, to find the action of $U_g$ on $O_{[J_1, J_2]}$, we only need to conjugate $O_{[J_1, J_2]}$ by the unitary operator

$$\lambda^{J_2 + \lfloor k/2\rfloor}(g, g^{-1})\lambda^{J_2 - 1 + \lfloor k/2\rfloor}(g, 1)\cdots\lambda^{J_1 + 1 - \lfloor(k+1)/2\rfloor}(g, 1)\lambda^{J_1 - \lfloor(k+1)/2\rfloor}(g, 1). \tag{40}$$

Here, $\lfloor x \rfloor$ is the greatest integer less than or equal to $x$. Using the assumption that the defect has width $k$, we find that the above unitary operator has support only on sites $J_1 - k \leq j \leq J_2 + k$. We conclude that the width of the symmetry operator $U_g$ is at most $k$.

## 2.4 Operators → defects

Now we establish the more difficult side of the correspondence. Namely, we construct an invertible topological defect of width at most $2k$ from a unitary symmetry operator $U_g$ of width $k$.

The basic idea is to truncate the symmetry operator $U_g$ to an operator that implements the symmetry in only half of the chain. More precisely, we want to find a unitary operator $U_g^{\leq J}$ such that for a local operator $O_j$ supported on site $j$ we have

$$U_g^{\leq J} O_j (U_g^{\leq J})^{-1} = \begin{cases} U_g O_j (U_g)^{-1}, & \text{for } j \leq J - k, \\ O_j, & \text{for } j > J + k. \end{cases} \tag{41}$$

If such a truncated symmetry operator[22] exists, we claim that a defect Hamiltonian is given by conjugating the untwisted Hamiltonian by this operator. Namely,

$$H_g^{(J,J+1)} \equiv U_g^{\leq J} H \left( U_g^{\leq J} \right)^{-1}. \tag{42}$$

Below, we first demonstrate that this equation does indeed define a localized defect, which is topological. Next, we will discuss the existence and construction of the truncated symmetry operator.

Before going to the next discussion, we want to emphasize that we only consider systems with a finite, but large, number of sites. For instance, in a more precise version of equation (42), we consider a truncated symmetry operator $U_g^{J,J'}$ that acts on a finite region between sites $J'$ and $J$, where $J' \ll J - 2k$. In that case, by conjugating the untwisted Hamiltonian with this truncated operator we in fact create a pair of topological defects. The additional defect is the inverse defect $g^{-1}$ and is localized around site $J'$. However, since the defects are localized, we can only focus on the region around $J$ to find the expression for a single defect. In what follows, in order to simplify the discussion, we will not specify the details of the system far away from the region of interest.

**Proof of locality:** To see that $H_g^{(J,J+1)}$ defines a defect that is localized near link $(J, J+1)$, we need to show that it differs from the untwisted Hamiltonian only near that link. Equivalently, we need to show that the difference, $H_g^{(J,J+1)} - H$, is a local operator supported near link $(J, J+1)$. The untwisted Hamiltonian is written as a sum of local terms, i.e.

$$H = \sum_j H_j, \tag{43}$$

where $H_j$ has support on at most sites $j, j+1, \cdots, j+l$ for some integer $l$. We claim that $H_g^{(J,J+1)} - H$ has support on at most sites $J - k - l < j \leq J + k + l$. Using (43) we find

$$H_g^{(J,J+1)} - H = \sum_{j \leq J+k} \left( U_g^{\leq J} H_j \left( U_g^{\leq J} \right)^{-1} - H_j \right) + \sum_{j > J+k} \left( U_g^{\leq J} H_j \left( U_g^{\leq J} \right)^{-1} - H_j \right). \tag{44}$$

Since the truncated symmetry operator commutes with operators supported on sites $j > J + k$, we find that the second term in the equation above vanishes. Therefore $H_g^{(J,J+1)} - H$ has

---

[22]Such a non-local operator, when it is not necessarily unitary, is called a 'disorder' operator [101], or a 'twist' field [102] in the continuum.

support on at most sites $j \leq J + k + l$. Using the fact that $U_g$ commutes with the untwisted Hamiltonian we find

$$H_g^{(J,J+1)} = \left(U_g^{>J}\right)^{-1} H \, U_g^{>J} \,, \tag{45}$$

where $U_g^{>J} = U_g (U_g^{\leq J})^{-1}$ is also a truncated symmetry operator that, because of (41), has support at most on sites $J > J - k$. Therefore we find that the operator

$$H_g^{(J,J+1)} - H = \sum_{j > J-k-l} \left(\left(U_g^{>J}\right)^{-1} H_j U_g^{>J} - H_j\right) + \sum_{j \leq J-k-l} \left(\left(U_g^{>J}\right)^{-1} H_j U_g^{>J} - H_j\right), \tag{46}$$

is at most supported on sites $j > J - k - l$. Thus we have proven that (42) defines a localized defect.

**Proof of the topological property:** To show that the defect is topological we construct the movement operators as

$$U_g^J \equiv U_g^{\leq J} \left(U_g^{\leq J-1}\right)^{-1} \,. \tag{47}$$

This operator is local and has support on at most sites $J - k \leq j \leq J + k$. This can be shown by using (41) and noting that the operator commutes with any local operator $O_j$ supported on site $j > J + k$ or $j > J - k$. Thus we find that the defect has a width of at most $2k$.

Above, we constructed a defect from the truncated symmetry operator. But does the truncated symmetry operator always exist? If we insist on keeping the Hilbert space of the defect the same as the untwisted Hilbert space, then such a truncated symmetry operator might not exist in general. However, a truncated symmetry operator always exists if we add local ancillas representing the defect degrees of freedom. Such a truncated operator is constructed in the context of quantum cellular automata (QCA) [103] (see also [84]). We review this construction below and use it to construct a defect Hamiltonian and defect Hilbert space. Moreover, we comment on when the truncated symmetry exists without having to add ancillas.

**Construction of the truncated symmetry**

To construct a truncated symmetry operator, first, we add a decoupled copy $B$ of the original system $A$ that has trivial dynamics. Namely, we consider the extended Hilbert space and Hamiltonian

$$\tilde{\mathcal{H}} = \mathcal{H} \otimes \mathcal{H} \,, \qquad \text{and} \qquad \tilde{H} = H \otimes \mathbb{1} \,. \tag{48}$$

The first factor describes the original system $A$, and the second factor is system $B$.

Next, we extend the original symmetry operator $U_g$ to

$$\tilde{U}_g = U_A U_B^{-1} \,, \tag{49}$$

where $U_A = U_g \otimes \mathbb{1}$ is the original symmetry operator acting on system $A$, and $U_B = \mathbb{1} \otimes U_g$. We will show below that $\tilde{U}_g$ can be truncated by using the *swap* operator

$$S = \prod_j s_j \,, \tag{50}$$

which exchanges system $A$ and $B$. The swap operator itself can be truncated since it is written as a product of local unitaries. The local swap operator $s_j$ is supported on site $j$ and satisfies

$$s_j \left(O_j \otimes \mathbb{1}\right) s_j^{-1} = \mathbb{1} \otimes O_j \,, \qquad \text{and} \qquad s_j^2 = 1 \,. \tag{51}$$

Using the swap operator, we rewrite the extended symmetry operator as

$$\tilde{U}_g = U_A U_B^{-1} = S U_B S U_B^{-1}$$
$$= \left( \prod_j s_j \right) \left( \prod_j U_B s_j U_B^{-1} \right). \tag{52}$$

This operator can be truncated to the unitary operator

$$\tilde{U}_g^{\leq J} \equiv \left( \prod_{j \leq J+k} s_j \right) \left( \prod_{j \leq J} U_B s_j U_B^{-1} \right), \tag{53}$$

which we now show is indeed a truncated symmetry in the sense of equation (41).

Using the fact that the symmetry operator $U_g$ has width $k$ we find

$$\tilde{U}_g^{\leq J} \left( O_j \otimes \mathbb{1} \right) \left( \tilde{U}_g^{\leq J} \right)^{-1} = \begin{cases} U_g O_j (U_g)^{-1} \otimes \mathbb{1}, & \text{for } j \leq J, \\ \mathbb{1} \otimes O_j, & \text{for } j = J+1, \cdots, J+k, \\ O_j \otimes \mathbb{1}, & \text{for } j > J+k, \end{cases} \tag{54}$$

and

$$\tilde{U}_g^{\leq J} \left( \mathbb{1} \otimes O_j \right) \left( \tilde{U}_g^{\leq J} \right)^{-1} = \begin{cases} \mathbb{1} \otimes (U_g)^{-1} O_j U_g, & \text{for } j \leq J-k, \\ \mathbb{1} \otimes O_j, & \text{for } j > J+k. \end{cases} \tag{55}$$

Therefore, $\tilde{U}_g^{\leq J}$ satisfy the truncated condition of equation (41).

**Defect Hilbert space:** As shown above, we can construct a defect Hamiltonian of width at most $2k$ as

$$H_g^{(J,J+1)} = \tilde{U}_g^{\leq J} \left( H \otimes \mathbb{1} \right) \left( \tilde{U}_g^{\leq J} \right)^{-1}. \tag{56}$$

However, the underlying Hilbert space $\mathcal{H} \otimes \mathcal{H}$ is too large. To find the true defect Hilbert space, we need to get rid of the decoupled degrees of freedom associated with system $B$. To do such, we project system $B$ into one of its states. We choose a *product state* $|0\rangle \in \mathcal{H}$, where $|0\rangle\langle 0| = \prod_j \pi_j$ and $\pi_j$ is a hermitian projection operator at site $j$.[23] The projector into this state is given by

$$\mathbb{1} \otimes \Pi \equiv \prod_j \mathbb{1} \otimes \pi_j. \tag{57}$$

For the system without a defect, imposing $\mathbb{1} \otimes \Pi = 1$ decouples system $B$ and gives us back the original system.

Now for the system with the defect, we consider the projection operator

$$\Pi^{(J,J+1)} = \tilde{U}_g^{\leq J} \left( \mathbb{1} \otimes \Pi \right) (\tilde{U}_g^{\leq J})^{-1}. \tag{58}$$

Note that $\Pi^{(J,J+1)}$ commutes with the defect Hamiltonian $H_g^{(J,J+1)}$ and is still written as a product of local commuting projectors. We define the defect Hilbert space $\mathcal{H}_g^{(J,J+1)}$ as the $\Pi^{(J,J+1)} = 1$ subspace of $\mathcal{H} \otimes \mathcal{H}$.[24] We emphasize that the defect Hilbert space only differs from the untwisted Hilbert space around site $J$. This is because the algebra of local operators away

---

[23]A product state exists in theories with a tenor product Hilbert space. In systems with additional local constraints, we can still consider such a product state by relaxing the local constraints of system $B$.

[24]The defect system is a local lattice system since $\Pi^{(J,J+1)}$ is a product of local commuting projectors. In other words, there is a local operator algebra generated by operators that commute with the local constraints $\tilde{U}_g^{\leq J} (\mathbb{1} \otimes \pi_j)(\tilde{U}_g^{\leq J})^{-1} = 1$. These local constraints are the analog of Gauss's law constraints in gauge theories.

from site $J$ is isomorphic for both the system with and without the defect. An isomorphism is given by the unitary operator $\tilde{U}_g^{\leq J}$.

In summary, we have constructed the defect Hamiltonian

$$H_g^{(J,J+1)} = \tilde{U}_g^{\leq J} \, (H \otimes \mathbb{1}) \, (\tilde{U}_g^{\leq J})^{-1} \,, \tag{59}$$

and defect Hilbert space

$$\mathcal{H}_g^{(J,J+1)} = \left\langle |\Psi\rangle \in \mathcal{H} \otimes \mathcal{H} \, \middle| \, \tilde{U}_g^{\leq J} \, (\mathbb{1} \otimes \Pi) \left(\tilde{U}_g^{\leq J}\right)^{-1} |\Psi\rangle = |\Psi\rangle \right\rangle \,. \tag{60}$$

To make sure there are no remaining decoupled degrees of freedom, we should check that there is no local operator commuting with $H_g^{(J,J+1)}$ and the constraint $\Pi^{(J,J+1)}$. A defect satisfying this condition is called a simple defect. If such a local operator exists, by conjugating it with $\tilde{U}_g^{\leq J}$, we find a local operator that commutes with both $H \otimes \mathbb{1}$ and $\mathbb{1} \otimes \Pi$. However, such an operator cannot exist because we are assuming that the original Hamiltonian does not commute with any local operator (see the paragraph about simple defects in Section 2.2). More generally, even if there are local operators that commute with the Hamiltonian, we consider a symmetric deformation of the Hamiltonian to eliminate such operators. This is always possible if the algebra of local operators has a trivial center or, equivalently, if there is no exact 1-form symmetry on the lattice.[25]

Let us consider an example that demonstrates the generality of the construction above. Below, we construct a topological defect for the lattice translation symmetry. This is an important example since the translation symmetry (in bosonic theories) can never be truncated in the original Hilbert space and we necessarily need to construct a non-trivial defect Hilbert space. In Appendix C, we discuss the case of antiunitary symmetries.

**Topological defect for translation:** Take the lattice translation operator $T$ acting as

$$T \, O_j \, T^{-1} = O_{j+1} \,. \tag{61}$$

We consider a translation invariant Hamiltonian, and without loss of generality, we assume that it has the following form

$$H = \sum_j H_j^0 H_{j+1}^1 \,. \tag{62}$$

Here, $H_j^0$ and $H_j^1$ are local operators supported on site $j$, where $T H_j^{0,1} T^{-1} = H_{j+1}^{0,1}$. We double the system and extend the translation operator to $\tilde{T} = T \otimes T^{-1}$. Note that the translation symmetry has width $k = 1$. Following our general construction, we truncate this symmetry operator to find

$$\tilde{T}_+^{\leq J} \equiv \left( \prod_{j \leq J+1} s_j \right) \left( \prod_{j \leq J} (\mathbb{1} \otimes T) s_j (\mathbb{1} \otimes T)^{-1} \right) \,. \tag{63}$$

The subscript $+$ here indicates that we are considering the translation to the right.

The action of the truncated symmetry on local operators is given by

$$\tilde{T}_+^{\leq J} \left(O_j \otimes \mathbb{1}\right) \left(\tilde{T}_+^{\leq J}\right)^{-1} = \begin{cases} O_{j+1} \otimes \mathbb{1}, & \text{for } j \leq J \,, \\ \mathbb{1} \otimes O_j, & \text{for } j = J+1 \,, \\ O_j \otimes \mathbb{1}, & \text{for } j \geq J+2 \,, \end{cases} \tag{64}$$

---

[25]An exact 1-form symmetry is defined by a local operator that is topological in spacetime [6]. This topological condition means that the operator commutes with any other local operator. Conversely, any operator in the center of the algebra of local operators commutes with any finite-range Hamiltonian, making it topological in spacetime.

and

$$\tilde{T}_+^{\leq J}\left(\mathbb{1}\otimes O_j\right)\left(\tilde{T}_+^{\leq J}\right)^{-1} = \begin{cases} \mathbb{1}\otimes O_{j-1}, & \text{for } j \leq J+1, \\ \mathbb{1}\otimes O_j, & \text{for } j \geq J+2. \end{cases} \tag{65}$$

Using these relations, we find the defect Hamiltonian

$$H_{T_+}^{(J,J+1)} = \sum_{j\leq J}\left(H_j^0 H_{j+1}^1 \otimes \mathbb{1}\right) + H_{J+1}^0 \otimes H_{J+1}^1 + H_{J+2}^1 \otimes H_{J+1}^0 + \sum_{j\geq J+2}\left(H_j^0 H_{j+1}^1 \otimes \mathbb{1}\right). \tag{66}$$

The defect Hilbert space is given by considering the projection operator

$$\Pi^{(J,J+1)} = \prod_{j\neq J+1} \mathbb{1}\otimes \pi_j. \tag{67}$$

This projection operator, projects out all the degrees of freedom of system $B$, besides its degree of freedom at site $J+1$. Doing this projection and denoting $H_B^{0,1} \equiv \mathbb{1}\otimes H_{J+1}^{0,1}$, we can rewrite the defect Hamiltonian as

$$H_{T_+}^{(J,J+1)} = \sum_{j\leq J}\left(H_j^0 H_{j+1}^1\right) + H_{J+1}^0 H_B^1 + H_B^0 H_{J+2}^1 + \sum_{j\geq J+2}\left(H_j^0 H_{j+1}^1\right). \tag{68}$$

This is almost the same as the untwisted system, except it has one extra site between sites $J+1$ and $J+2$.

Similarly, we can consider a defect for the inverse translation symmetry $T_- = T^{-1}$. The corresponding truncated symmetry operator acts as

$$\tilde{T}_-^{\leq J}\left(O_j\otimes \mathbb{1}\right)\left(\tilde{T}_-^{\leq J}\right)^{-1} = \begin{cases} O_{j-1}\otimes \mathbb{1}, & \text{for } j \leq J, \\ \mathbb{1}\otimes O_j, & \text{for } j = J+1, \\ O_j\otimes \mathbb{1}, & \text{for } j \geq J+2, \end{cases} \tag{69}$$

and

$$\tilde{T}_-^{\leq J}\left(\mathbb{1}\otimes O_j\right)\left(\tilde{T}_-^{\leq J}\right)^{-1} = \begin{cases} \mathbb{1}\otimes O_{j+1}, & \text{for } j \leq J-1, \\ O_j\otimes \mathbb{1}, & \text{for } j = J, J+1, \\ \mathbb{1}\otimes O_j, & \text{for } j \geq J+2. \end{cases} \tag{70}$$

The projection operator is

$$\Pi^{(J,J+1)} = (\pi_J \pi_{J+1}\otimes \mathbb{1}) \prod_{j\neq J+1} \mathbb{1}\otimes \pi_j. \tag{71}$$

Imposing $\Pi^{(J,J+1)} = 1$, effectively, projects out system $B$ completely and also the site $J$ of the original system. More precisely, it also exchanges the site $J+1$ of the two systems. We undo the latter by conjugation with the local unitary $s_{J+1}$. In the end, we find the defect Hamiltonian

$$H_{T_-}^{(J,J+1)} = \sum_{j\leq J-2}\left(H_j^0 H_{j+1}^1\right) + H_{J-1}^0 H_{J+1}^1 + \sum_{j\geq J+1}\left(H_j^0 H_{j+1}^1\right). \tag{72}$$

In summary, we find that the defect for translation symmetries $T$ and $T^{-1}$ correspond, respectively, to adding an extra site and removing one site of the chain. These topological defects are inverse of each other and fuse to the trivial defect.

**Finite-depth quantum circuits:** The problem of whether a symmetry operator of finite width can be truncated or not is a well-known problem in the context of QCA. This problem is completely settled in 1+1d systems with a Hilbert space that is a tensor product of *finite-dimensional* local Hilbert spaces [104]. In that case, a symmetry operator of finite width is truncatable, i.e. it is a finite-depth quantum circuit, if it corresponds to an internal symmetry. In particular, the translation symmetry operator cannot be truncated.[26]

Another example where the symmetry operator might not be truncatable, is when the Hilbert space does not have a tensor product structure. For instance, in gauge theories, the Hilbert space is given by imposing local Gauss's law constraints. The simplest example is the 1+1d lattice $\mathbb{Z}_2$ gauge theory. The Hamiltonian is identically zero. The Hilbert space is given by a tensor product of local two-dimensional Hilbert spaces with the Gauss's law constraints

$$\sigma^z_j \sigma^z_{j+1} = 1, \qquad \text{for all } j. \tag{73}$$

This theory has two ground states that are related by a $\mathbb{Z}_2$ symmetry generated by the Wilson line operator

$$W = \prod_j \sigma^x_j. \tag{74}$$

Because of Gauss's law constraint, the truncated symmetry operator is not gauge invariant and does not act on the untwisted Hilbert space. However, we can create a defect at link $(J, J+1)$ by changing Gauss's law at that link to $\sigma^z_J \sigma^z_{J+1} = -1$. We want to point out that this defect is *not* simple since there exists a local operator $\sigma^z_j$ that commutes with the Hamiltonian. An open question is whether, with the extra assumption of simplicity, all internal symmetry operators are truncatable in 1+1d even in systems without a tensor product Hilbert space.

We end this section by remarking that the truncatability of symmetry operators has also been discussed in the context of continuum field theory. In particular, the authors of [105] define global symmetry as splittable if the corresponding symmetry operator can be truncated. Note that even when the symmetry operator is not splittable, there still exist movement operators that move the defect in space and map one defect Hilbert space to another one. We conclude that any global symmetry has a generalized split property, where the movement operator is considered a generalization of the truncated symmetry operator.

# 3 Anomalies and gauging on the lattice

In this section, we derive our formula for 1+1d 't Hooft anomalies on the lattice. We introduce a method for gauging arbitrary internal symmetries on the lattice and show the anomaly is the obstruction to gauging. By imposing spatial symmetries, we find mixed anomalies that capture the obstruction to gauging while preserving the spatial symmetries. In particular, we compute the LSM anomaly capturing the mixed anomaly between an internal symmetry and lattice translation.

## 3.1 The anomaly

We compute the anomaly locally using the unitary fusion operators, defined in equation (30), which describes the fusion of topological defects. The basic idea is to perform F-moves that probe the associativity of the fusion operation.[27] We consider three defects $g_1, g_2, g_3$ on three

---

[26]We thank Wilbur Shirley for the discussions on this and related points.

[27]The notion of F-move in physics goes back to the study of 1+1d rational conformal field theory [106]. It was originally introduced in the context of the representation theory of SU(2), where it was called Racah coefficients or the 6j-symbols.

consecutive links and fuse them in two different orders. The result must differ by a phase $F^j(g_1, g_2, g_2) \in U(1)$ that captures the anomaly (as we show later) and is given by

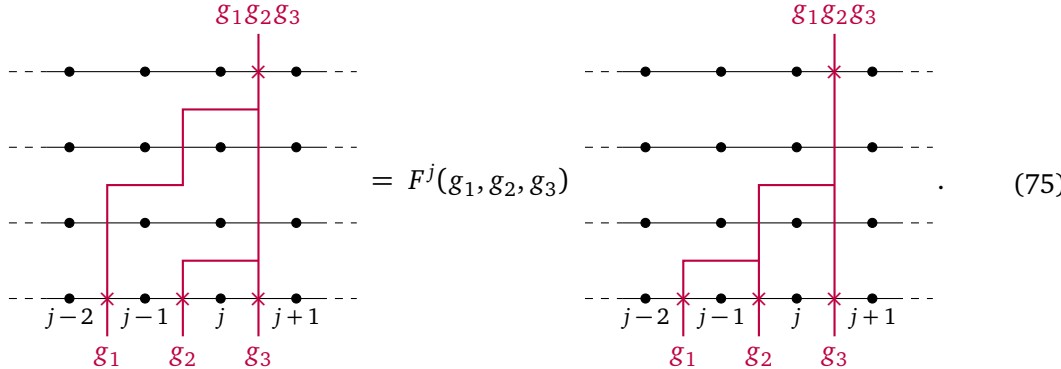

$$= F^j(g_1, g_2, g_3) \qquad . \qquad (75)$$

The spacetime diagram on the left and right-hand sides correspond to unitary local operators conjugating the Hamiltonian of three defects $g_1, g_2, g_3$ to the one with a single defect $g_1 g_2 g_3$ on link $(j, j+1)$. Therefore their ratio, $F^j(g_1, g_2, g_3)$, commutes with the defect Hamiltonian $H^{(j,j+1)}_{g_1 g_2 g_3}$. Since we assume the defects are simple,[28] there is no non-trivial local operator commuting with the defect Hamiltonian. Thereby, we find that $F^j(g_1, g_2, g_3)$ is just a phase factor.

The F-moves define a map $F^j : G \times G \times G \to U(1)$, which in terms of the fusion operators is written as

$$\lambda^j(g_1, g_2 g_3) \lambda^{j-1}(g_1, 1) \lambda^j(g_2, g_3) = F^j(g_1, g_2, g_3) \lambda^j(g_1 g_2, g_3) \lambda^{j-1}(g_1, g_2). \qquad (76)$$

Since the fusion operators are well-defined up to a phase, $F^j$ has some ambiguities that we discuss below. Moreover, the map $F^j$ satisfies a modified pentagon identity. These two facts are the defining properties of the map $F^j$, which we show to be the 't Hooft anomaly.

**I. Modified pentagon identity:** We insert four defects $g_1, g_2, g_3, g_4$ on consecutive links and consider the fusion operators that fuse them in two different orders: $g_1 \times (g_2 \times (g_3 \times g_4))$ and $((g_1 \times g_2) \times g_3) \times g_4$. F-moves relate these fusion operators in two different ways, which leads to a constraint on the map $F^j$.

---

[28]According to this assumption, either there is no local operator that commutes with the Hamiltonian, or the Hamiltonian can be symmetrically deformed to eliminate such operators. This assumption is equivalent to assuming that the algebra of local operators has a trivial center, or in other words, there is no exact 1-form symmetry.

The first series of F-moves is

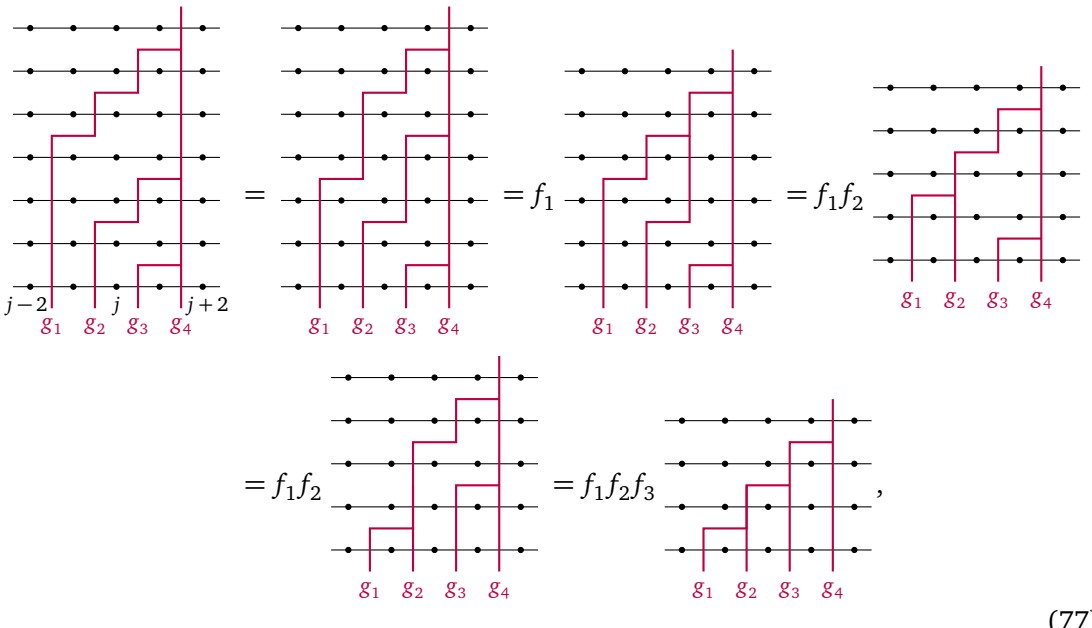

$$(77)$$

where $f_1 = F^{j+1}(g_1, g_2, g_3g_4)$, $f_2 = F^j(g_1, g_2, 1)$, and $f_3 = F^{j+1}(g_1g_2, g_3, g_4)$. The first and fourth equalities above follow from the assumption that the defects have width $k = 1$. Namely, $\lambda^{j-1}(g_1, 1)$ commutes with $\lambda^{j+1}(g_2, g_3g_4)$, and $\lambda^{j-1}(g_1, g_2)$ commutes with $\lambda^{j+1}(g_3, g_4)$.

The other sequence of F-moves is

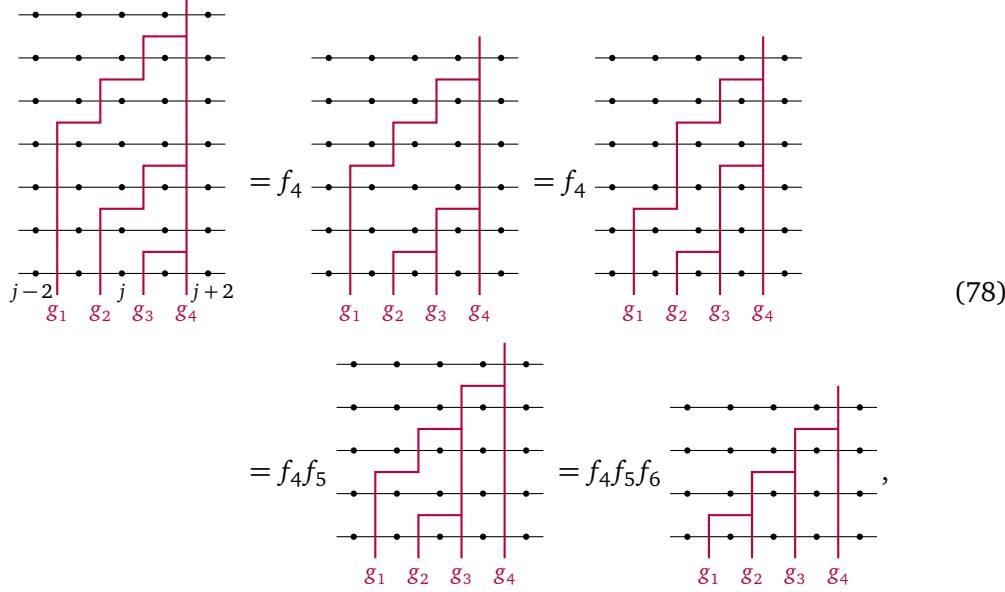

$$(78)$$

where $f_4 = F^{j+1}(g_2, g_3, g_4)$, $f_5 = F^{j+1}(g_1, g_2g_3, g_4)$, and $f_6 = F^j(g_1, g_2, g_3)$. The second equality above holds since $\lambda^{j-1}(g_1, 1)$ commutes with $\lambda^{j+1}(g_2g_3, g_4)$.

We find $f_1f_2f_3 = f_4f_5f_6$, which is equivalent to

**Modified pentagon equation**

$$\frac{F^j(g_2, g_3, g_4) \, F^j(g_1, g_2g_3, g_4) \, F^{j-1}(g_1, g_2, g_3)}{F^j(g_1g_2, g_3, g_4) \, F^j(g_1, g_2, g_3g_4)} = F^{j-1}(g_1, g_2, 1). \qquad (79)$$

This is the lattice version of the pentagon equation in the continuum where the term on the right-hand side is absent.[29]

**II. Phase ambiguity:**  Consider a phase redefinition of the fusion operators given by a map $\gamma^j : G \times G \to U(1)$, which changes

$$\lambda^j(g_1, g_2), \qquad \text{into} \qquad \gamma^j(g_1, g_2)\lambda^j(g_1, g_2). \tag{80}$$

Such a phase ambiguity leads to the identification

$$F^j(g_1, g_2, g_3) \sim F^j(g_1, g_2, g_3)\frac{\gamma^j(g_2, g_3)\gamma^j(g_1, g_2 g_3)\gamma^{j-1}(g_1, 1)}{\gamma^j(g_1 g_2, g_3)\gamma^{j-1}(g_1, g_2)}. \tag{81}$$

**Relation to group cohomology**

We denote the equivalence class of the map $F^j$ under the identification (81) by $[F^j]$. In the next subsection, we will show that $[F^j]$ is the obstruction to gauging $G$ and thus is the 't Hooft anomaly in $G$. Before doing that, we want to show that our result is consistent with the proposed group cohomology classification of anomalies. Specifically, we identify $[F^j]$ as an element of $H^3(G, U(1))$. Moreover, imposing additional (spatial) symmetries restricts the possible phase ambiguities in (80). As a result, we find less identification on $F^j$ and thus more possibilities for the anomaly, specifically the mixed 't Hooft anomaly between $G$ and the additional imposed symmetry. In particular, imposing lattice translation symmetry identifies $[F^j]$ as an element of $H^3(G, U(1)) \times H^2(G, U(1))$.

**Anomaly cocycles:**  We decompose $F^j : G \times G \times G \to U(1)$ into the *3-cocycle* $\omega^j$ and *2-cocycle* $\alpha^j$,[30] which are denoted by

$$\omega^j(g_1, g_2, g_3) \equiv \frac{F^j(g_1, g_2, g_3)}{F^j(g_1, g_2, 1)}, \qquad \text{and} \qquad \alpha^j(g_1, g_2) \equiv F^j(g_1, g_2, 1). \tag{82}$$

Below, we rewrite equations (79) and (81) in terms of these cocycles and phrase them in the language of group cohomology. See [21] for a physicist review of group cohomology theory.

The modified pentagon equation (79) is equivalent to the following two conditions on the cocycles. We find $\omega^j$ satisfies the 3-cocycle condition

$$(\delta_3 \omega^j)(g_1, g_2, g_3, g_4) \equiv \frac{\omega^j(g_2, g_3, g_4)\omega^j(g_1, g_2 g_3, g_4)\omega^j(g_1, g_2, g_3)}{\omega^j(g_1 g_2, g_3, g_4)\omega^j(g_1, g_2, g_3 g_4)} = 1, \tag{83}$$

and $\alpha^j$ satisfies the condition

$$(\delta_2 \alpha^j)(g_1, g_2, g_2) \equiv \frac{\alpha^j(g_2, g_3)\alpha^j(g_1, g_2 g_3)}{\alpha^j(g_1 g_2, g_3)\alpha^j(g_1, g_2)} = \frac{\omega^j(g_1, g_2, g_3)}{\omega^{j-1}(g_1, g_2, g_3)}, \tag{84}$$

which reduces to the standard 2-cocycle condition if $\omega^j = \omega^{j-1}$. Note that equation (84) is given by setting $g_4 = 1$ in equation (79), and (83) is given by taking the ratio of (79) with

---

[29]The appearance of the extra term on the right-hand side is due to the fusion operations occurring at different sites. It is possible to implement the fusions at a single site and find the standard pentagon equation, as described in [78]. The advantage of the current method is that our F-symbols capture both the anomaly in $G$ and the mixed anomaly between $G$ and lattice translation.

[30]The 2-cochain $\alpha^j$ satisfies the 2-cocycle condition when there is a translation symmetry. We loosely use the term "2-cocycle" to refer to it, even without a translation symmetry.

respect to (84). Therefore, the two conditions above are equivalent to the modified pentagon equation.

Let us also decompose the map $\gamma^j : G \times G \to U(1)$, that redefines the phase of fusion operators, into

$$\gamma_2^j(g_1, g_2) \equiv \frac{\gamma^j(g_1, g_2)}{\gamma^j(g_1, 1)}, \qquad \text{and} \qquad \gamma_1^j(g_1) \equiv \gamma^j(g_1, 1). \tag{85}$$

In terms of these maps, the identification (81) is equivalent to the following identifications. We find that $\omega^j$ is well-defined up to

$$\omega^j(g_1, g_2, g_3) \sim \omega^j(g_1, g_2, g_3) \frac{\gamma_2^j(g_2, g_3) \gamma_2^j(g_1, g_2 g_3)}{\gamma_2^j(g_1 g_2, g_3) \gamma_2^j(g_1, g_2)}. \tag{86}$$

The extra term on the right-hand side is known as a coboundary/exact term and is denoted by $(\delta_2 \gamma_2^j)(g_1, g_2, g_3)$. This identifies the equivalence class of $\omega^j$, denoted by $[\omega^j]$, as an element of $H^3(G, U(1))$. For the 2-cocycle $\alpha^j$ we find

$$\alpha^j(g_1, g_2) \sim \alpha^j(g_1, g_2) \frac{\gamma_1^j(g_2) \gamma_1^j(g_1)}{\gamma_1^j(g_1 g_2)} \frac{\gamma_2^j(g_1, g_2)}{\gamma_2^{j-1}(g_1, g_2)}, \tag{87}$$

which is equivalent to $[\alpha^j] \in H^2(G, U(1))$ when $\gamma_2^j = \gamma_2^{j-1}$.

Putting everything together we find the constraints

$$\delta_3 \omega^j = 1, \qquad \text{and} \qquad \delta_2 \alpha^j = \frac{\omega^j}{\omega^{j-1}}, \tag{88}$$

and identifications

$$\omega^j \sim \omega^j (\delta_2 \gamma_2^j), \qquad \text{and} \qquad \alpha^j \sim \alpha^j (\delta_1 \gamma_1^j) \frac{\gamma_2^j}{\gamma_2^{j-1}}. \tag{89}$$

Depending on whether we impose a lattice translation symmetry or not we find the following classification for the anomaly:

a. **No translation symmetry:** If we do not impose any other symmetry besides $G$, the phase redefinitions given by $\gamma^j$ are arbitrary. As a result, by doing the phase redefinition

$$\gamma^j = \begin{cases} \left( \alpha^j \alpha^{j-1} \cdots \alpha^1 \right)^{-1}, & \text{for } j \geq 1, \\ \alpha^0 \alpha^{-1} \cdots \alpha^{j+1}, & \text{for } j \leq 0, \end{cases} \tag{90}$$

the identification (89) becomes

$$\omega^j \sim \omega^0, \qquad \text{and} \qquad \alpha^j \sim 1, \tag{91}$$

for all $j$. Here we have used the fact that $\delta_2 \alpha^j = \omega^j / \omega^{j-1}$ and $\alpha^j(g, 1) = 1$. This means that when there are no other symmetries we can completely eliminate the 2-cocycle $\alpha^j$. Thus, the anomaly involving only $G$ is given by $[\omega] \equiv [\omega^0] \in H^3(G, U(1))$.

b. **With translation symmetry:** Imposing a lattice translation symmetry $T$ satisfying $T \lambda^j T^{-1} = \lambda^{j+1}$, restricts the phase redefinitions and the cocycles to be $j$ independent. Therefore, we can drop the superscript $j$ on these cocycles and find the constraints

$$\delta_3 \omega = 1, \qquad \text{and} \qquad \delta_2 \alpha = 1, \tag{92}$$

and identifications

$$\omega \sim \omega(\delta_2\gamma_2), \qquad \text{and} \qquad \alpha \sim \alpha(\delta_1\gamma_1). \tag{93}$$

This means that the anomaly involving $G$ and the translation symmetry is given by

$$[\omega] \in H^3(G, U(1)), \qquad \text{and} \qquad [\alpha] \in H^2(G, U(1)). \tag{94}$$

We end the discussion here by remarking that for continuous symmetries, the cocycles that appeared above are restricted and should satisfy some continuity constraints. Since anomalies in the continuous case are conjecturally classified by the Borel cohomology group [21], a conjecture is that one only uses Borel measurable cocycles.

Our working assumption, which may or may not lead to Borel measurable cocycles, is the following. Consider a connected Lie group $G$ and divide the group manifold of $G$ into a finite number of patches. We assume that there exists a choice of patches such that the defect Hilbert spaces inside each patch are constant and the defect Hamiltonian is a continuous function of the group parameters inside each patch. For the case of $G = U(1)$ that is discussed in Section 4.3, we choose a fundamental domain $[0, 2\pi)$ to parametrize the group elements. This is equivalent to covering $U(1)$ by two patches where the defect Hamiltonian is continuous inside each patch.

**Reflection symmetry:**  Let us briefly comment on the mixed-anomaly involving $G$ and a reflection symmetry R acting as $\mathsf{R}O_j\mathsf{R}^{-1} = O_{-j}$. For simplicity, we assume the reflection symmetry commute with the internal symmetry $G$. Analogous to the case of translation symmetry, preserving reflection symmetry requires imposing the relation

$$\mathsf{R}\lambda^j(g_1, g_2)\mathsf{R}^{-1} = \lambda^{-j}(g_2^{-1}, g_1^{-1}). \tag{95}$$

Moreover, one must consider only those phase redefinitions, $\gamma^j$, that respect this relation. Then, a mixed-anomaly in R and $G$ corresponds to the impossibility of satisfying the above relation while also having $F^j = 1$. From the discussion below, it becomes clear that these conditions are required to gauge $G$ while keeping the reflection symmetry operator, R, gauge-invariant.

## 3.2  Gauging (non-on-site) symmetries

Here, we describe a method for gauging internal symmetries, which may not be on-site, using topological defects. We show that the anomalies discussed above are the only obstructions to this gauging procedure. Our method generalizes the standard gauging of on-site symmetries and consists of two steps: adding dynamical gauge "fields" and imposing Gauss's law. In the first step, we enlarge the Hilbert space by adding all possible defects on links. In the second step, we impose Gauss's law constraints on sites and find that it is consistent to do so only if the anomaly vanishes.

**1) Adding dynamical gauge fields:**  Consider an internal symmetry $G$ and sum over the insertion of $G$ topological defects on each link. Concretely, consider a periodic chain with $L$ sites and extend its Hilbert space $\mathcal{H}$ to

$$\widetilde{\mathcal{H}} \equiv \bigoplus_{g_1,\dots,g_L \in G} \mathcal{H}_{g_1;g_2;\dots;g_L} \sim \bigoplus_{g_1,\dots,g_L \in G} \underset{1 \quad 2 \quad 3 \qquad L-1 \quad L \quad 1}{\overset{g_1 \quad g_2 \qquad\qquad g_{L-1} \quad g_L}{\bullet\!-\!\times\!-\!\bullet\!-\!\times\!-\!\bullet\ -\ -\ -\ \bullet\!-\!\times\!-\!\bullet\!-\!\times\!-\!\bullet}} , \tag{96}$$

where $\mathcal{H}_{g_1;g_2;\dots;g_L}$ is the defect Hilbert space for the system with $g_j$ defect on link $(j, j+1)$. When $G$ is a discrete group, this corresponds to adding $|G|$-dimensional Hilbert spaces on links,

and for continuous symmetries we add scalar fields on links for each generator of $G$. We extend the Hamiltonian $H$ to

$$\widetilde{H} \equiv \sum_{g_1,\ldots,g_L \in G} H_{g_1;g_2;\ldots;g_L} \otimes \Pi_{g_1,g_2,\ldots,g_L}, \tag{97}$$

where $H_{g_1;g_2;\ldots;g_L}$ is the defect Hamiltonian for the system with $g_j$ defect on link $(j,j+1)$, and $\Pi_{g_1,g_2,\ldots,g_L}$ is the projection operator into $\mathcal{H}_{g_1;g_2;\ldots;g_L}$.

For simplicity, we consider the case where $G$ is discrete and each defect Hilbert space is an identical copy of the untwisted Hilbert space. In that case, we find the extended Hilbert space and Hamiltonian

$$\widetilde{\mathcal{H}} = \mathcal{H} \bigotimes_{j=1}^{L} \mathbb{C}^{|G|}, \qquad \text{and} \qquad \widetilde{H} = \sum_{g_1,\ldots,g_L} H_{g_1;\ldots;g_L} \otimes |g_1,\ldots,g_L\rangle\langle g_1,\ldots,g_L|_{\text{links}}. \tag{98}$$

Here, $|g_1,\ldots,g_L\rangle\langle g_1,\ldots,g_L|_{\text{links}} = \bigotimes_{j=1}^{L} |g_j\rangle\langle g_j|_{(j,j+1)}$ where $|g_j\rangle\langle g_j|_{(j,j+1)}$ is a projection operator acting on link $(j,j+1)$ that fixes the value of the defect on that link to be $g_j \in G$.

**2) Imposing Gauss's law:**   Now we extend the fusion operators $\lambda^J(g,h)$ into

$$\widetilde{\lambda}^J(g,h) \equiv \lambda^J(g,h) \otimes \left(|1\rangle\langle g|_{(J-1,J)} \otimes |gh\rangle\langle h|_{(J,J+1)}\right) \bigotimes_{j \neq J-1,J} \mathbb{1}_{(j,j+1)}, \tag{99}$$

which acts non-trivially only on sites $J$ and $J+1$ and links $(J-1,J)$ and $(J,J+1)$. Crucially, the extended fusion operator $\widetilde{\lambda}^J(g,h)$ commutes with the extended Hamiltonian $\widetilde{H}$. This follows from the defining property of the fusion operators, that is

$$\lambda^J(g_{j-1},g_j)H_{g_1;\ldots;g_L} = H_{g_1;\ldots;g_{j-2};1;g_{j-1}g_j;g_{j+1};\ldots;g_L}\lambda^J(g_{j-1},g_j). \tag{100}$$

We note that $\widetilde{\lambda}^J(g,h)$ is similar to a gauge transformation, except that it is not unitary.

We construct unitary operators out of the fusion operators by considering

$$\mathcal{G}^j(g) \equiv \sum_{a,b \in G} \left(\widetilde{\lambda}^j(ag^{-1},gb)\right)^\dagger \widetilde{\lambda}^j(a,b) \sim \sum_{a,b \in G} \quad , \tag{101}$$

which acts unitarily on both the degrees of freedom on sites and links. The unitary operators $\mathcal{G}^j(g)$ generate a local $G$ symmetry of the extended Hamiltonian since we have

$$\mathcal{G}^j(g)\mathcal{G}^j(h) = \mathcal{G}^j(gh), \qquad \text{and} \qquad \mathcal{G}^j(g)^\dagger = \mathcal{G}^j(g^{-1}), \tag{102}$$

for $g,h \in G$. The global part of this symmetry is

$$\widetilde{U}_g = \mathcal{G}^L(g)\mathcal{G}^{L-1}(g)\cdots\mathcal{G}^2(g)\mathcal{G}^1(g). \tag{103}$$

By restricting to the untwisted sector, we recover the global symmetry of the unextended system, i.e. $U_g = {}_{\text{links}}\langle 1,\ldots,1|\widetilde{U}_g|1,\ldots,1\rangle_{\text{links}}$. This can be seen by comparing equations (33) and (34) with (101).

Finally, to *gauge* this local symmetry we need to impose the Gauss's law constraints $\mathcal{G}^j(g) = 1$ at all sites. The Gauss's law constraint at site $j$ is given by

$$P_j \equiv \frac{1}{|G|} \sum_{g \in G} \mathcal{G}^j(g) = 1, \tag{104}$$

where $P_j$ is a local projection operator.

All in all, the gauged theory is described by the lattice Hamiltonian $\tilde{H}$ and Hilbert space $\tilde{\mathcal{H}}$ that is subject to the Gauss's law constraints (104). However, it is consistent to impose Gauss's law constraints if they commute with each other at different sites. As we will show below, this is the case if and only if $F^j(g_1, g_2, g_3) = 1$ for all $g_1, g_2, g_3 \in G$.

Since we are assuming that the defects have a width of $k = 1$, the Gauss's law constraints that are at least two sites apart necessarily commute with each other. Hence, we only need to check whether $P_j$ commutes with $P_{j-1}$ or not, which is equivalent to

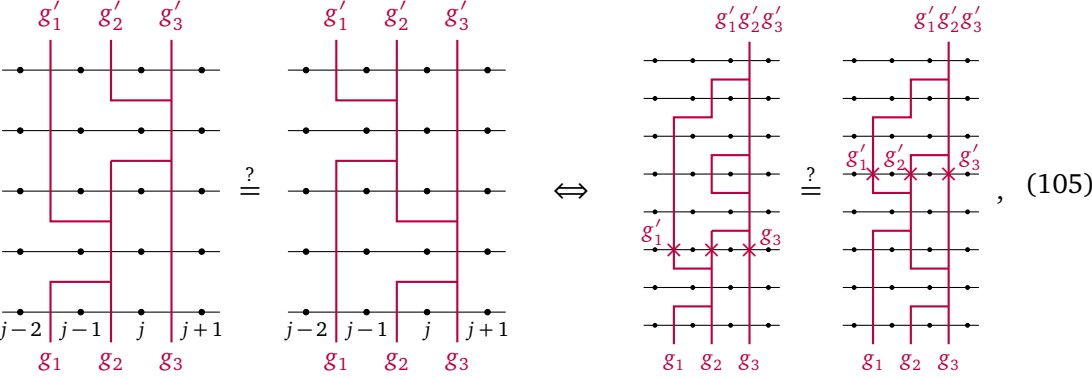

$$(105)$$

for all $g_1, g_2, g_3, g_1', g_2', g_3' \in G$ such that $g_1 g_2 g_3 = g_1' g_2' g_3'$. The two equations above are related by a unitary operator implementing the fusion $g_1' \times (g_2' \times g_3')$ on the top. Now we use F-moves to simplify the equation on the right. First, we use the following to eliminate the "bubbles"

$$\Leftrightarrow \qquad \lambda^j(g,h)\big(\lambda^j(g,h)\big)^\dagger = 1. \qquad (106)$$

Next, we perform F-moves on both sides of the equation to get

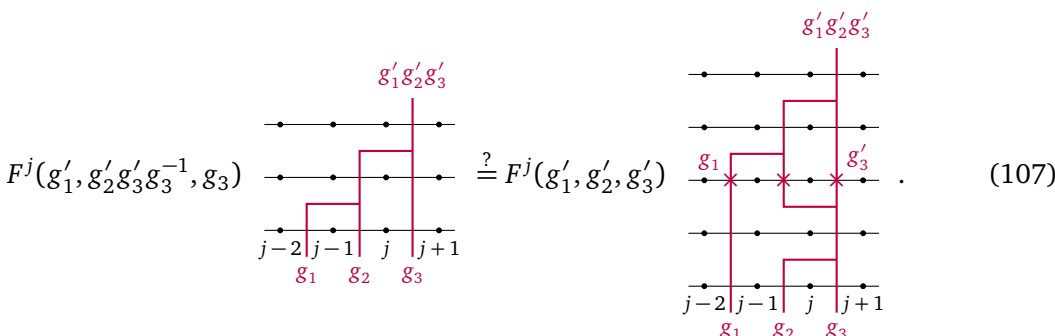

$$F^j(g_1', g_2'g_3'g_3^{-1}, g_3) \quad \overset{?}{=} \quad F^j(g_1', g_2', g_3') \qquad\qquad . \qquad (107)$$

Finally, by applying two additional F-moves on the right-hand side, we simplify the equation further and find that

$$P_j \text{ commutes with } P_{j-1} \quad \Leftrightarrow \quad \frac{F^j(g_1', g_2'g_3'g_3^{-1}, g_3)}{F^j(g_1, g_2, g_3)} = \frac{F^j(g_1', g_2', g_3')}{F^j(g_1, g_1^{-1}g_1'g_2', g_3')}. \qquad (108)$$

By setting $g_1' = 1$, $g_2' = g_1$, and $g_3' = g_2 g_3$ the equation above reduces to $F^j(g_1, g_2, g_3) = 1$. This is because $F^j(1, g, h) = F^j(g, 1, h) = 1$, which follows from $\lambda^j(1, g) = 1$.

In summary, we find that Gauss's law constraints commute with each other at different sites if and only if $F^j = 1$. However, even if $F^j$ is not trivial, we might be able to trivialize it using the phase redefinition of the fusion operators reflected in the identification (81). Therefore, we justify that $[F^j]$ is the 't Hooft anomaly since it is the obstruction to gauging.

As mentioned before, when we allow arbitrary phase redefinition, the anomaly $[F^j]$ is identified with the cohomology class $[\omega] \in H^3(G, U(1))$. By imposing additional symmetries, we have less freedom in phase redefinitions. For instance, by imposing a lattice translation symmetry $T$, we only allow for $j$-independent phase redefinition that respect the relation $T\lambda^j T^{-1} = \lambda^{j+1}$. The latter relation leads to the condition $TP_j T^{-1} = P_{j+1}$, that is the statement that the translation operator is gauge invariant. Therefore, the (generalized) LSM anomaly $[\alpha] \in H^2(G, U(1))$, which we find by restricting to $j$-independent phase redefinitions, is the obstruction to gauging $G$ while preserving the translation symmetry. Thus, we justify that the LSM anomaly is the mixed 't Hooft anomaly between $G$ and lattice translation symmetry.[31]

Throughout the discussion in this section, we assumed that the defects have width $k = 1$. The discussion can be generalized for $k > 1$ to compute the $H^3$ and LSM anomalies involving $T^l$ for any $l \geq k$. To do such, we simply insert the defects so they are always $l$ sites apart. In this way, we can compute the LSM anomaly involving subgroups of translation generated by $T^l$. Consider the subgroups generated by $T^k$ and $T^{k+1}$. These two subgroups generate all the translations; therefore, knowing the anomaly for these two subgroups is enough to determine the anomaly in the full translation symmetry group $\mathbb{Z}$. Therefore, our method can be generalized to compute the anomaly for defects of arbitrary finite width.

Relatedly, we can consider defects for lattice translations and "almost" compute a 3-cocycle in $\mathbb{Z} \times G$.[32] As mentioned before, defects for lattice translations correspond to adding or removing a site. Removing $k$ sites is a topological defect of width $k$, and adding $k$ sites is a defect of width 1. Thus we can consider defects for adding an arbitrary number of sites. This way, we compute a 3-cocycle in the semi-group $G \times \mathbb{Z}^{\geq 0}$. By doing the computation, we find that this 3-cocycle is equal to

$$\omega(g_1 T^{n_1}, g_2 T^{n_2}, g_3 T^{n_3}) = \omega(g_1, g_2, g_3)(\alpha(g_1, g_2))^{n_3}, \qquad \text{for } n_1, n_2, n_3 \in \mathbb{Z}^{\geq 0}. \tag{109}$$

However, knowing the 3-cocycle for non-negative integers is as good as knowing the full 3-cocycle.

## 3.3 LSM-type constraints

Here, we discuss generalized LSM-type constraints that can be viewed as a version of 't Hooft anomaly matching on the lattice. We argue that a theory with a non-trivial anomaly cannot have a unique gapped ground state. Below, we do not attempt to provide a rigorous proof and only outline an idea for one.[33]

Since the anomaly takes discrete values, it cannot change under continuous deformations. Specifically, the anomaly is an element of the group cohomology $H^3(G, U(1)) \times H^2(G, U(1))$, which is a discrete group. Therefore, a theory with a non-trivial anomaly does not have a symmetric deformation to a gapped phase with a unique ground state. This statement on the lattice simply follows from the fact that the anomaly is a property of the symmetry operator rather than the Hamiltonian.

To demonstrate that $[F^j] \neq 1$ forbids a unique gapped ground state, we first need the following lemma: If the untwisted Hamiltonian has a unique *gapped* ground state, then the

---

[31]Since translation symmetry does not act internally, we do not know how to gauge translation symmetry and identify the mixed anomaly as the obstruction to gauging translation while preserving $G$.

[32]Because of the Künneth formula $H^2(G, U(1)) \times H^3(G, U(1)) \cong H^3(\mathbb{Z} \times G, U(1))$, the anomalies that we have computed can be packaged as a 3-cocycle in $\mathbb{Z} \times G$.

[33]A rigorous proof was recently presented in [107].

defect Hamiltonians must also have a unique gapped ground state. Specifically, for a gapped Hamiltonian, the ground states of the Hamiltonian with and without a defect are related by a local unitary operator. To show this, let us consider the unitary operation of creating a pair of defects, $g$ and $g^{-1}$, and separating them far apart using the movement operator. Assume we create them at site 1 and move one of them to site $J \gg 1$. This process is described by a sequence of unitary operators that implements the symmetry action on the interval $[1, J]$. In a gapped system, the action of such a unitary operator on the ground state $|\Omega\rangle$ is equivalent to acting with a unitary operator with support only around sites 1 and $J$. In other words

$$\lambda^J(g,1) \cdots \lambda^2(g,1) \left(\lambda^1(g^{-1},g)\right)^{-1} |\Omega\rangle = u_J u_1 |\Omega\rangle, \tag{110}$$

where $u_1$ and $u_J$ are localized around sites 1 and $J$, respectively. This follows from the *split property*[34] of the *gapped* ground state $|\Omega\rangle$.[35,36] Therefore, we conclude that for a gapped state, the topological defect of $g$ can be created locally by the action of local unitary operator $u_j$. We note that this is strictly true in the infinite volume limit. In finite volume, the operator $u_j$ only approximately creates the defect. Therefore, we find that the ground states of the system with and without a defect are related by local unitary operators. In particular, if the untwisted problem has a unique gapped ground state then the twisted problem must also have a unique gapped ground state.

We denote the ground state of the defect Hamiltonian $H_g^{(j,j+1)}$ by $\left|\Omega_g^{(j,j+1)}\right\rangle$ and call it a *defect state*. Crucially, the defect states only differ from each other around the location of the defect. In other words, the expectation value of a local operator inserted far away from the defect should be the same for two different defect states. This follows from the split property of the ground state $|\Omega\rangle$, which allows us to construct the ground state $\left|\Omega_{g;h}^{(j-k,j-k+1);(j,j+1)}\right\rangle$ of the system with two defects, that are separated by $k \gg 1$ sites, from the ground state of the system with one defect. We take $k$ to be much larger than the correlation length of the system which specifies the characteristic width of the defect in defect states.

By acting with the fusion operators, we map these ground states to each other and find

$$\lambda^j(g,h)\lambda^{j-1}(g,1) \cdots \lambda^{j-k+1}(g,1)\left|\Omega_{g;h}^{(j-k,j-k+1);(j,j+1)}\right\rangle = \varphi^j(g,h)\left|\Omega_{gh}^{(j,j+1)}\right\rangle, \tag{111}$$

for some phase factors $\varphi^j(g,h) \in U(1)$. Here, we take $k$ to be large enough such that the phase $\varphi^j(g,h)$ is independent of other defects inserted $k$ sites away from the interval $[j-k,j]$. The idea is to compute the anomaly in terms of these phases and show that it is trivial in cohomology. Consider the defect state $\left|\Omega_{g_1;g_2;g_3}^{(j-2k,j-2k+1);(j-k,j-k+1);(j,j+1)}\right\rangle$ and apply F-moves to find

$$\frac{\varphi^j(g_2,g_3)\varphi^j(g_1,g_2g_3)\varphi^{j-k}(g_1,1)}{\varphi^j(g_1g_2,g_3)\varphi^{j-k}(g_1,g_2)} = \omega^j(g_1,g_2,g_3)\left(\alpha^j(g_1,g_2)\right)^k. \tag{112}$$

This implies that the anomaly $\omega^j$ is trivial in cohomology.

Now let us consider the case with a lattice translation symmetry. We impose a translation symmetry $T$ and choose the defect states such that

$$T^k\left|\Omega_g^{(j,j+1)}\right\rangle = \left|\Omega_g^{(j+k,j+k+1)}\right\rangle. \tag{113}$$

---

[34]The split property, proven in [108], follows from the area law of entanglement entropy, which holds for the ground states of gapped Hamiltonians in 1+1d [109].

[35]The split property states that the Hilbert space of a gapped system in infinite volume is factorized. Therefore, cutting the space into halves and considering the action of the symmetry on one half of the space must be realized as a local unitary $u_1$ on the state. See, for instance, [107, Lemma 4.1].

[36]The split property was used in [40, 41] to prove a generalized LSM theorem rigorously for on-site symmetries. See also [110, 111] for rigorous results on the classification of 1+1d and 2+1d SPTs.

This implies the phases that appear in equation (111) are $j$-independent, i.e. $\varphi^j = \varphi^{j+k}$. Therefore, we find that the anomaly $(\alpha^j(g_1, g_2))^k$ is trivial in cohomology. Repeating the same argument for $k+1$ we find that both $(\alpha)^k$ and $(\alpha)^{k+1}$ are trivial, and therefore the LSM anomaly $[\alpha]$ must be trivial. This is a generalized LSM theorem stating that a non-trivial generalized LSM anomaly $[\alpha] \in H^2(G, U(1))$ forbids a unique gapped ground state.

**The continuum limit and emanant symmetries**

Here, following [57], we explain how the anomaly of the UV lattice system is matched with the continuum theory in the IR. We give a geometric argument that F-move, and therefore $F^j(g_1, g_2, g_3)$, captures the anomaly in the continuum theory. For lattice translation symmetry, the anomaly is matched by an *emanant* symmetry in the IR as explained in [57]. The emanant symmetry is an internal symmetry of the IR theory that is *generated* by the continuum limit of the lattice translation by one site.

The basic idea is the following relation between topological defects and background gauge fields. Coupling a quantum field theory to flat background gauge fields for an internal symmetry $G$ corresponds to inserting $G$-symmetry defects across various cycles in both the time direction and space [6]. Given a configuration of topological defects, we can compute the holonomy of the gauge fields as follows. Note that the holonomies around various 1-cycles capture all the gauge invariant information of flat background gauge fields. For a 1-cycle $\gamma$, each time it intersects with a $g$-symmetry defect, it contributes a $g$-holonomy. Thus the total holonomy is given by the product of the group elements associated with the defects that the 1-cycle intersects. This determines a flat $G$ connection given a network of topological defects inserted in the spacetime manifolds. The opposite procedure, however, is not unique and this is where anomalies appear.

An internal symmetry $G$ is anomaly free if one can unambiguously assign a configuration of topological defects for a given flat $G$ connection (see e.g. [6]). To describe a $G$ connection, we divide the spacetime manifold into different patches and assign $G$ transition functions across different patches. We insert topological defects on the interface between different patches that match the transition functions. In this way, we have coupled the theory to flat background gauge fields. However, dividing the manifold into patches is not unique and different choices are related by gauge transformations. Any two configurations that describe the same flat $G$ connection are related by a sequence of F-moves (75). Another ambiguity in describing flat connections in terms of topological defects is the phase ambiguity associated with junctions in the triple intersection of topological defects, i.e. the fusion operators. Therefore, the theory is anomaly-free if one can redefine the fusion operators' phases such that $F^j = 1$.

The argument above explains why the anomaly $F^j$ involving an internal symmetry $G$ matches with the IR theory in the continuum. This is because the phases $F^j$ have a discrete classification and thereby cannot change under taking the continuum limit. The argument applies to mixed anomalies between $G$ and any other symmetry $H$, such as lattice translations. We note that imposing $H$ restricts the possible phase redefinition of the junction operators. We require that the configuration of $G$-defects transform covariantly with respect to $H$. Phase redefinition of the junctions corresponds to local counterterms in the continuum. A mixed anomaly between $G$ and $H$ arises if one cannot trivialize the anomalous phase $F^j$ by adding counterterms that respect the symmetry $H$. When $H$ is a translation symmetry, the mixed anomaly is captured by a 2-cocycle $[\alpha] \in H^2(G, U(1))$.

# 4 Examples and other spacetime symmetries

Here, we will study four examples demonstrating the general results in the previous sections. In Section 4.4, we generalize our results to the case of fermionic theories and consider a fermionic example.

## 4.1 LSM for on-site symmetries – The Heisenberg chain

Here we consider internal symmetries that act, either linearly or projectively, on-site. Specifically, we consider systems with a $G$ internal symmetry such that the symmetry operators are written as

$$U_g = \prod_j U_g^j, \tag{114}$$

where $U_g^j$ is a unitary operator that acts nontrivially only on site $j$. Moreover, these local operators form a representation of $G$ in the sense that

$$U_g^j U_h^j \propto U_{gh}^j. \tag{115}$$

The symmetry is realized linearly on-site if there exists a phase redefinition of the operators $U_g^j$ such that $U_g^j U_h^j = U_{gh}^j$. Otherwise, the symmetry is realized projectively on-site.

We also assume that there exists a translation symmetry $T$ satisfying

$$T U_g^j T^{-1} = U_g^{j+1}. \tag{116}$$

This condition implies that the internal symmetry $G$ commutes with the translation symmetry, i.e. $TU_g = U_g T$. Later, we comment on the case where this is not the case.

**Heisenberg chain:** An example is the anti-ferromagnetic spin-half Heisenberg chain, which appears in the original LSM theorem. The Hilbert space is a tensor product of spin-$\frac{1}{2}$ degrees of freedom per each site of the chain. The Hamiltonian is

$$H = \frac{1}{2} \sum_j \left( \sigma_j^x \sigma_{j+1}^x + \sigma_j^y \sigma_{j+1}^y + \sigma_j^z \sigma_{j+1}^z \right), \tag{117}$$

where $\vec{\sigma}_j = (\sigma_j^x, \sigma_j^y, \sigma_j^z)$ are the Pauli matrices acting on site $j$. This model has an internal $SO(3)$ symmetry that is realized projectively on-site. Each symmetry operator is written as a product of local factors

$$U_{\vec{\theta}}^j = \exp\left( i\vec{\theta} \cdot \frac{\vec{\sigma}_j}{2} \right), \tag{118}$$

where $\vec{\theta} = (\theta_x, \theta_y, \theta_z)$ parametrizes the $SO(3)$ transformations. The spin-half degrees of freedom on each site form a projective representation of $SO(3)$. The system also has a translation symmetry $T$ satisfying

$$T \vec{\sigma}_j T^{-1} = \vec{\sigma}_{j+1}. \tag{119}$$

We now show that in such systems, there is no anomaly in the internal symmetry but there is a generalized LSM anomaly. The case where the internal symmetry commutes with translation symmetry has been considered in [41].

**The anomaly**

Since the symmetry operators are written as a product of local factors, the fusion operators are given as

$$\lambda^j(g,h) = U_g^j. \tag{120}$$

This relation is independent of the Hamiltonian. Now, recall that the anomaly is given by the relation

$$\lambda^j(g_1, g_2 g_3)\lambda^{j-1}(g_1, 1)\lambda^j(g_2, g_3) = F^j(g_1, g_2, g_3)\lambda^j(g_1 g_2, g_3)\lambda^{j-1}(g_1, g_2). \tag{121}$$

Using this we find $U_{g_1}^j U_{g_2}^j = F^j(g_1, g_2, g_3)U_{g_1 g_2}^j$.

We find that $F^j(g_1, g_2, g_3)$ is independent of $g_3$, therefore $\omega^j(g_1, g_2, g_3) = 1$ and hence there is no anomaly involving only $G$. Furthermore, because of the translation symmetry $F^j(g_1, g_2, g_3)$ is $j$-independent. Therefore $\alpha(g_1, g_2) = F^j(g_1, g_2, 1)$ where

$$U_{g_1}^j U_{g_2}^j = \alpha(g_1, g_2) U_{g_1 g_2}^j. \tag{122}$$

Thus we see that the LSM anomaly $[\alpha] \in H^2(G, U(1))$ determines the projective representation of the symmetry on each site.

**Dipole-type symmetries:**   Above, we only considered the cases where the internal symmetry commutes with lattice translation. Let us now consider the case that there is a semidirect product structure between the internal and translation symmetry. In such cases, there exists an automorphism $\phi : G \to G$ such that

$$T U_g^j T^{-1} = U_{\phi(g)}^{j+1}. \tag{123}$$

An example of this is the dipole symmetry [112].

We claim that even in such cases the projective phases above correspond to an anomaly. We note that $\alpha^j$ is no longer $j$-independent. Instead, we have the relation

$$\alpha^j(g_1, g_2) = \alpha^{j+1}(\phi(g_1), \phi(g_2)). \tag{124}$$

In such cases, $[\alpha^1]$ can be identified as the anomaly in the following sense. We can gauge $G$ while preserving the translation symmetry only if $[\alpha^1] = 1$.

We note that preserving the translation symmetry means that after gauging $G$ the translation operator $T$ must be gauge invariant and hence commutes with the Gauss's law constraints. Moreover, the relation (123), leads to the fact that $T P_j T^{-1} = P_{j+1}$. Thus we preserve the translation symmetry if we only consider $j$-independent phase redefinitions of the fusion operators that do not violate the relation (123). Therefore, we see that $F^j$ can be trivialized by $j$-independent phase redefinitions if and only if $[\alpha^1] = 1$. Therefore $[\alpha^1] \in H^2(G, U(1))$ is the anomaly involving $G$ and lattice translation.

## 4.2   Non-invertible lattice translation – A gauged XYZ chain

As shown above, if an internal symmetry $G$ is realized projectively on-site, it can be gauged, but it has a mixed anomaly with the lattice translation symmetry. What happens to the translation symmetry if we gauge the internal symmetry with an LSM anomaly? As we have argued, gauging the internal symmetry necessarily violates the original translation symmetry since it is no longer gauge invariant. We claim that the translation symmetry does not disappear completely after gauging $G$; instead, it appears as a *non-invertible* lattice translation symmetry. Here we demonstrate this in the case of the XYZ chain. We left the discussion of the more general cases elsewhere.

**The XYZ chain**

Consider a deformation of the Heisenberg chain such that it preserves the $\mathbb{Z}_2 \times \mathbb{Z}_2$ subgroup of the $SO(3)$ symmetry. The Hamiltonian is

$$H = \frac{1}{2}\sum_j \left( J_x \sigma_j^x \sigma_{j+1}^x + J_y \sigma_j^y \sigma_{j+1}^y + J_z \sigma_j^z \sigma_{j+1}^z \right), \tag{125}$$

for coupling constants $J_x, J_y, J_z \in \mathbb{R}$. There is a $\mathbb{Z}_2^X \times \mathbb{Z}_2^Y$ symmetry generated by

$$X = \prod_j \sigma_j^x, \qquad \text{and} \qquad Y = \prod_j \sigma_j^y. \tag{126}$$

We note that the $\mathbb{Z}_2^X \times \mathbb{Z}_2^Y$ has an LSM anomaly since it is realized projectively on-site. The projective phase can be seen from the commutation relation $\sigma_j^x \sigma_j^y = -\sigma_j^y \sigma_j^x$. The fusion operators are given by

$$\lambda^j (X^{\epsilon_1} Y^{\nu_1}, X^{\epsilon_2} Y^{\nu_2}) = (\sigma_j^x)^{\epsilon_1}(\sigma_j^y)^{\nu_1} = \qquad , \tag{127}$$

where $\epsilon_1, \nu_1, \epsilon_2, \nu_2 \in \{0,1\}$ parametrize the group $\mathbb{Z}_2^X \times \mathbb{Z}_2^Y = \{1, X, Y, XY\}$. Using the formula for the anomaly, we find the anomaly cocycles:

$$\alpha(X^{\epsilon_1} Y^{\nu_1}, X^{\epsilon_2} Y^{\nu_2}) = (-1)^{\nu_1 \epsilon_2}, \qquad \text{and} \qquad \omega = 1. \tag{128}$$

**Gauging $\mathbb{Z}_2^X \times \mathbb{Z}_2^Y$**

The phases given by $\alpha$ cannot be trivialized by $j$-independent phase redefinitions and thus is a true mixed anomaly between $\mathbb{Z}_2^X \times \mathbb{Z}_2^Y$ and lattice translation. However, we can trivialize it by using $j$-dependent phases at the expense of violating the translation symmetry.

We consider the $j$-dependent phase redefinition $\gamma^j (X^{\epsilon_1} Y^{\nu_1}, X^{\epsilon_2} Y^{\nu_2}) = (-1)^{j \nu_1 \epsilon_2}$, and find the new fusion operators

$$\lambda^j (X^{\epsilon_1} Y^{\nu_1}, X^{\epsilon_2} Y^{\nu_2}) = (-1)^{j \nu_1 \epsilon_2} (\sigma_j^x)^{\epsilon_1}(\sigma_j^y)^{\nu_1}. \tag{129}$$

Using equation (81), we find that the above phase redefinition indeed trivializes the anomalous phase $\alpha$, i.e.

$$F^j (X^{\epsilon_1} Y^{\nu_1}, X^{\epsilon_2} Y^{\nu_2}, X^{\epsilon_3} Y^{\nu_3}) = (-1)^{\nu_1 \epsilon_2} \frac{(-1)^{j \nu_2 \epsilon_3}(-1)^{j \nu_1 (\epsilon_2 + \epsilon_3)}}{(-1)^{j(\nu_1 + \nu_2)\epsilon_3}(-1)^{(j-1)\nu_1 \epsilon_2}} = 1. \tag{130}$$

Now we perform the gauging explicitly, following the procedure outlined in Section 3.2.

**Adding gauge fields:** First, we extend the system by adding dynamical gauge fields on links. We put the system on a periodic chain with an even number of sites $L$ to ensure that the phase redefinition denoted above is well-defined, i.e. $\lambda^{j+L} = \lambda^j$. We add two qubits on each link associated with the $\mathbb{Z}_2^X$ and $\mathbb{Z}_2^Y$ symmetries, and denote them by

$$|X^\epsilon Y^\nu\rangle_{(j,j+1)} = |\epsilon\rangle_{X_{(j,j+1)}} \otimes |\nu\rangle_{Y_{(j,j+1)}}, \qquad \text{for } \epsilon, \nu \in \{0,1\}. \tag{131}$$

Here $\epsilon = 1$ (and $\nu = 1$) corresponds to a non-trivial $\mathbb{Z}_2^X$ (and $\mathbb{Z}_2^Y$) defect on link $(j, j+1)$. We define the corresponding Pauli operators by $\mathbf{X}_{j,j+1}^x, \mathbf{X}_{j,j+1}^y, \mathbf{X}_{j,j+1}^z$ and $\mathbf{Y}_{j,j+1}^x, \mathbf{Y}_{j,j+1}^y, \mathbf{Y}_{j,j+1}^z$.[37]

The extended Hamiltonian is given by

$$\widetilde{H} = \frac{1}{2} \sum_{j=1}^{L} \Big( J_x \sigma_j^x \, \mathbf{Y}_{j,j+1}^z \, \sigma_{j+1}^x + J_y \sigma_j^y \, \mathbf{X}_{j,j+1}^z \, \sigma_{j+1}^y + J_z \sigma_j^z \, \mathbf{X}_{j,j+1}^z \mathbf{Y}_{j,j+1}^z \, \sigma_{j+1}^z \Big), \qquad (132)$$

where $L = 2\tilde{L}$. To see this, we note that the defect Hamiltonian for a $\mathbb{Z}_2^X$ defect on link $(j, j+1)$ corresponds to changing the sign of the terms $\sigma_j^y \sigma_{j+1}^y$ and $\sigma_j^z \sigma_{j+1}^z$ in the Hamiltonian. Similarly, for the $\mathbb{Z}_2^Y$ defect we change the sign of $\sigma_j^x \sigma_{j+1}^x$ and $\sigma_j^z \sigma_{j+1}^z$.

**Gauss's law:** Next, we find the Gauss's law constraints. Using equation (101), we find

$$
\begin{aligned}
\mathcal{G}^j(X^\epsilon Y^\nu) &= \sum_{\epsilon_1, \nu_1, \epsilon_2, \nu_2} \lambda^j \big( X^{\epsilon_1 - \epsilon} Y^{\nu_1 - \nu}, X^{\epsilon_2 + \epsilon} Y^{\nu_2 + \nu} \big)^\dagger \lambda^j \big( X^{\epsilon_1} Y^{\nu_1}, X^{\epsilon_2} Y^{\nu_2} \big) \\
&\quad \otimes \big| X^{\epsilon_1 - \epsilon} Y^{\nu_1 - \nu}, X^{\epsilon_2 + \epsilon} Y^{\nu_2 + \nu} \big\rangle \big\langle X^{\epsilon_1} Y^{\nu_1}, X^{\epsilon_2} Y^{\nu_2} \big|_{(j-1,j);(j,j+1)} \\
&= \sum_{\epsilon_1, \nu_1, \epsilon_2, \nu_2} (-1)^{j(\nu_1 - \nu)(\epsilon_2 + \epsilon) + j\nu_1 \epsilon_2} (\sigma_j^y)^{\nu_1 - \nu} (\sigma_j^x)^{\epsilon_1 - \epsilon} (\sigma_j^x)^{\epsilon_1} (\sigma_j^y)^{\nu_1} \\
&\quad \otimes \big| X^{\epsilon_1 - \epsilon} Y^{\nu_1 - \nu} \big\rangle \big\langle X^{\epsilon_1} Y^{\nu_1} \big|_{(j-1,j)} \otimes \big| X^{\epsilon_2 + \epsilon} Y^{\nu_2 + \nu} \big\rangle \big\langle X^{\epsilon_2} Y^{\nu_2} \big|_{(j,j+1)}.
\end{aligned}
\tag{133}
$$

The term that acts on sites can be simplified to $(-1)^{j(\nu_1 \epsilon + \epsilon_2 \nu + \epsilon \nu)} (-1)^{\epsilon(\nu_1 - \nu)} (\sigma_j^x)^\epsilon (\sigma_j^y)^\nu$. Therefore, we find

$$
\begin{aligned}
\mathcal{G}^j(X^\epsilon Y^\nu) &= (\sigma_j^x)^\epsilon (\sigma_j^y)^\nu \otimes \sum_{\epsilon_1} |\epsilon_1 - \epsilon\rangle\langle \epsilon_1|_{X_{(j-1,j)}} \otimes \sum_{\nu_1} (-1)^{(j+1)(\nu_1 + \nu)\epsilon} |\nu_1 - \nu\rangle\langle \nu_1|_{Y_{(j-1,j)}} \\
&\quad \otimes \sum_{\epsilon_2} (-1)^{j\epsilon_2 \nu} |\epsilon_2 + \epsilon\rangle\langle \epsilon_2|_{X_{(j,j+1)}} \otimes \sum_{\nu_2} |\nu_2 + \nu\rangle\langle \nu_2|_{Y_{(j,j+1)}} \\
&= (\sigma_j^x)^\epsilon (\sigma_j^y)^\nu (\mathbf{X}_{j-1,j}^x)^\epsilon (\mathbf{Y}_{j-1,j}^z)^{\epsilon(j+1)} (\mathbf{Y}_{j-1,j}^x)^\nu (\mathbf{X}_{j,j+1}^x)^\epsilon (\mathbf{X}_{j,j+1}^z)^{\nu j} (\mathbf{Y}_{j,j+1}^x)^\nu \\
&= \Big( (\mathbf{Y}_{j-1,j}^z)^{j+1} \mathbf{X}_{j-1,j}^x \sigma_j^x \mathbf{X}_{j,j+1}^x \Big)^\epsilon \Big( \mathbf{Y}_{j-1,j}^x \sigma_j^y \mathbf{Y}_{j,j+1}^x (\mathbf{X}_{j,j+1}^z)^j \Big)^\nu.
\end{aligned}
$$

In summary, the generators of the *local* $\mathbb{Z}_2^X \times \mathbb{Z}_2^Y$ symmetry of the extended Hamiltonian (132) are given by:

$$
\begin{aligned}
\mathcal{G}_j(X) &= (\mathbf{Y}_{j-1,j}^z)^{j+1} \mathbf{X}_{j-1,j}^x \sigma_j^x \mathbf{X}_{j,j+1}^x, \\
\mathcal{G}_j(Y) &= \mathbf{Y}_{j-1,j}^x \sigma_j^y \mathbf{Y}_{j,j+1}^x (\mathbf{X}_{j,j+1}^z)^j.
\end{aligned}
\tag{134}
$$

The terms with explicit $j$-dependence correspond to the $j$-dependent phase redefinition defined above. Their purpose is to make these local symmetries commute with each other at different sites. Finally, the Gauss's law at site $j$ is given by $\mathcal{G}_j(X) = 1$ and $\mathcal{G}_j(Y) = 1$.

After gauging and imposing Gauss's law constraints, the lattice translation symmetry $T$ is violated since it is not gauge invariant, i.e. $T\mathcal{G}_j T^{-1} \neq \mathcal{G}_{j+1}$. However, translation by two sites is preserved since $T^2 \mathcal{G}_j T^{-2} = \mathcal{G}_{j+2}$.

**Gauge fixing:** We can perform local unitary transformations to find a presentation where the Hilbert space is a tensor product of local factors with no Gauss's constraints. First, we use the unitary operators $\mathcal{G}_j(Y)$ to rotate the spins at sites such that $\sigma_j^x = 1$ for all $j$. Next, we use $\mathcal{G}_j(X)$ to set $\mathbf{Y}_{j-1,j}^x = 1$ for even $j$, and set $\mathbf{X}_{j,j+1}^z = 1$ for odd $j$. Doing these transformations amount to the following choice of gauge:

$$\sigma_j^x = 1, \quad \mathbf{Y}_{2l-1,2l}^x = 1, \quad \mathbf{X}_{2l-1,2l}^z = 1. \tag{135}$$

---

[37]Specifically, $\mathbf{X}_{j,j+1}^z = |0\rangle\langle 0|_{X_{(j,j+1)}} - |1\rangle\langle 1|_{X_{(j,j+1)}}$ and $\mathbf{Y}_{j,j+1}^z = |0\rangle\langle 0|_{Y_{(j,j+1)}} - |1\rangle\langle 1|_{Y_{(j,j+1)}}$.

The remaining degrees of freedom reside on even links. More precisely, the list of gauge-invariant operators are[38]

$$\mu_l^z = \sigma_{2l}^y \mathbf{X}_{2l,2l+1}^z \sigma_{2l+1}^y, \qquad \mu_l^x = \mathbf{X}_{2l,2l+1}^x,$$
$$\tau_l^z = \sigma_{2l}^x \mathbf{Y}_{2l,2l+1}^z \sigma_{2l+1}^x, \qquad \tau_l^x = \mathbf{Y}_{2l,2l+1}^x. \tag{136}$$

In terms of these operators, we view the lattice as a chain with $\tilde{L} = L/2$ sites. Using the Gauss's law constraints $\mathcal{G}_j(X) = \mathcal{G}_j(Y) = 1$, we write the Hamiltonian (132) of the gauged theory in terms of the new variables as

$$\widetilde{H} = \frac{1}{2} \sum_{l=1}^{\tilde{L}} \Big( J_x \left( \tau_l^z + \mu_l^x \mu_{l+1}^x \right) + J_y \left( \mu_l^z + \tau_l^x \tau_{l+1}^x \right) - J_z \left( \tau_l^z + \mu_l^x \mu_{l+1}^x \right) \left( \mu_l^z + \tau_l^x \tau_{l+1}^x \right) \Big), \tag{137}$$

where the Hilbert space consists of two qubits per site that are acted upon by $\vec{\mu}_l$ and $\vec{\tau}_l$.

**Non-invertible translation symmetry**

As mentioned above, the original lattice translation operator $T$ is violated since it is not gauge invariant. However, it appears as a non-invertible lattice translation symmetry. The continuum version of this phenomenon is discussed in [113, 114].[39] To find this non-invertible symmetry, it is useful to first discuss another symmetry of the system. Gauging the $\mathbb{Z}_2^X \times \mathbb{Z}_2^Y$ symmetry, leads to a *dual* $\mathbb{Z}_2^{\tilde{X}} \times \mathbb{Z}_2^{\tilde{Y}}$ symmetry generated by the Wilson line operators

$$\tilde{X} = \prod_j \mu_j^z, \qquad \text{and} \qquad \tilde{Y} = \prod_j \tau_j^z. \tag{138}$$

The translation symmetry $T$ is not gauge invariant since it does not preserve Gauss's law constraints. We notice that violations of Gauss's law correspond to the insertion of the dual symmetry defects. Specifically, the defects $\mathcal{G}_{2l}(X) = -1$ and $\mathcal{G}_{2l}(Y) = -1$, respectively, correspond to $\mathbb{Z}_2^{\tilde{X}}$ and $\mathbb{Z}_2^{\tilde{Y}}$ symmetry twists at site $l$ of the new chain. This is because $\mathbf{X}_{2l,2l+1}^z \mathbf{X}_{2l+1,2l+2}^z = \mathbf{Y}_{2l-1,2l}^x (\mu_l^z) \mathbf{Y}_{2l+1,2l+2}^z$ and $\mathbf{Y}_{2l,2l+1}^z \mathbf{Y}_{2l+1,2l+2}^z = \mathbf{X}_{2l-1,2l}^x (\tau_l^z) \mathbf{X}_{2l+1,2l+2}^x$ are the movement operators for these defects, and at the same time, generate the $\mathbb{Z}_2^{\tilde{X}} \times \mathbb{Z}_2^{\tilde{Y}}$ symmetry.

Therefore, the translation symmetry $T$ can be extended to an operator that acts on an extended Hilbert space with four sectors corresponding to the various $\mathbb{Z}_2^{\tilde{X}} \times \mathbb{Z}_2^{\tilde{Y}}$ twists. This is the hallmark of non-invertible symmetries; see [117] for a discussion in the continuum, and see [118] for an alternative perspective. Putting the theory on a closed chain and projecting onto the untwisted sector we find a non-invertible translation $\tilde{T}$ satisfying the algebra

$$\tilde{T} \times \tilde{T} = T^2 (1 + \tilde{X})(1 + \tilde{Y}). \tag{139}$$

Another way to see the non-invertible symmetry in this model is the following. By setting $J_z = 0$, we identify the system after gauging as the stack of two transverse-field Ising models. The $\vec{\mu}_l$ and $\vec{\tau}_l$ degrees of freedom, respectively, correspond to Ising models with magnetic field strengths given by $J_y/J_x$ and $J_x/J_y$. The Hamiltonian (137) is self-dual under gauging $\mathbb{Z}_2^{\tilde{X}} \times \mathbb{Z}_2^{\tilde{Y}}$ since gauging inverts the coupling constants $J_y/J_x$ and $J_x/J_y$, which can be undone by swapping $\vec{\mu}_l$ with $\vec{\tau}_l$. Because of the self-duality under gauging, the model has a non-invertible symmetry generated by the corresponding "duality defect" [8, 88].

---

[38]We thank Nati Seiberg for a discussion related to this point.

[39]See [58, 115, 116], for a discussion of detecting mixed anomalies on the lattice by gauging anomaly-free subgroups.

Finally, the non-invertible symmetry $\tilde{T}$ acts on operators as

$$
\begin{aligned}
\mu^z_{j-1} &\mapsto \tau^x_{j-1}\tau^x_j, & \mu^x_j &\mapsto \tau^z_j\,\tau^z_{j-1}\,\tau^z_{j-2}\,\tau^z_{j-3}\,\tau^z_{j-4}\cdots, \\
\tau^z_{j-1} &\mapsto \mu^x_{j-1}\mu^x_j, & \tau^x_j &\mapsto \mu^z_j\,\mu^z_{j-1}\,\mu^z_{j-2}\,\mu^z_{j-3}\,\mu^z_{j-4}\cdots.
\end{aligned}
\tag{140}
$$

The action above is only a formal, but practically useful, expression on infinite chains. On a closed chain, $\tilde{T}$ annihilate any state which is not invariant under the $\mathbb{Z}_2^{\tilde{X}} \times \mathbb{Z}_2^{\tilde{Y}}$ symmetry. The commutation relations between $\tilde{T}$ and $\mathbb{Z}_2^{\tilde{X}} \times \mathbb{Z}_2^{\tilde{Y}}$ invariant operators are given by

$$
\tilde{T}\,\mu^z_j = \tau^x_j\tau^x_{j+1}\,\tilde{T}\,, \qquad \text{and} \qquad \tilde{T}\,\mu^x_j\mu^x_{j+1} = \tau^z_{j+1}\,\tilde{T}\,.
\tag{141}
$$

### 4.3 Lattice chiral gauge theories – The modified Villain model

Here we discuss the modified Villain model of compact boson theory [119–121], which realizes anomalous chiral $U(1)_\mathrm{L}$ and $U(1)_\mathrm{R}$ symmetries of the compact boson theory on the lattice. We study the Hamiltonian description of this model that was presented in [57,122]. At the end of this section, we explain how to construct Abelian chiral gauge theories on the lattice using the modified Villain theory.

**The modified Villain XY-model**

The Hilbert space consists of non-compact fields $\phi_j$ at site $j$, and $\mathbb{Z}$-valued gauge fields $n_{j,j+1}$ on link $(j, j+1)$. We also have the conjugate momenta $p_j$ and compact electric-fields $E_{j,j+1}$ that satisfy the commutation relations

$$
[\phi_j, p_{j'}] = [n_{j,j+1}, E_{j',j'+1}] = i\delta_{j,j'}\,.
\tag{142}
$$

The Hamiltonian is

$$
H_{\mathrm{mV}} = \sum_j \left( \frac{1}{2R^2}p_j^2 + \frac{R^2}{2}\left( \frac{\phi_{j+1} - \phi_j}{2\pi} - n_{j,j+1} \right)^2 \right),
\tag{143}
$$

where the physical Hilbert space is given by imposing Gauss's law constraints

$$
G_j = e^{i(E_{j,j+1} - E_{j-1,j}) - 2\pi i p_j} = 1\,,
\tag{144}
$$

at each site $j$.

This model has a $U(1)_m \times U(1)_w$ symmetry generated by the conserved charges

$$
Q_m = \sum_j p_j\,, \qquad \text{and} \qquad Q_w = -\sum_j n_{j,j+1}\,.
\tag{145}
$$

One interesting feature of this model is its exact T-duality exchanging the degrees of freedom on sites with the degrees of freedom on links. The T-duality transformation maps gauge-invariant operators into gauge-invariant operators and hence respects Gauss's laws. However, the duality transformation does not commute with the gauge transformation generated by $G_j$. It only maps the gauge-invariant states $|\psi\rangle$, with $G_j|\psi\rangle = |\psi\rangle$, to themselves. But it does not preserve other eigenvalues of $G_j$.

We note that other eigenvalues of $G_j$ correspond to the insertion of winding-symmetry defects at site $j$. Moreover, since the duality transformation exchanges the momentum and winding symmetry, it should also exchange the corresponding topological defects. We slightly change the presentation of the model to make the T-duality more manifest.

**Alternative presentation**

In a slight rewriting of the model, we make the electric fields $E_{j,j+1}$ non-compact and impose their compactness by extra Gauss's-law type constraint that is $e^{2\pi i n_{j,j+1}} = 1$. We emphasize that this rewriting does not change the system. We interpret the new Gauss's law as coming from "magnetic" $\mathbb{Z}$ gauge transformations. This formulation has the advantage that different eigenvalues of the new Gauss's law correspond to the insertion of momentum-symmetry defects. We will see that in this formulation, T-duality manifestly exchanges the momentum and winding defects. In particular, the Hilbert space without any defect (i.e. the gauge invariant Hilbert space), is mapped into itself under the action of T-duality.

In the new presentation, we have a chain with non-compact fields $\phi_j$ and $\tilde{\phi}_{j,j+1}$ respectively at site $j$ and link $(j, j+1)$. The conjugate momenta are $p_j$ and $\tilde{p}_{j,j+1}$ that satisfy the commutation relations

$$[\phi_j, p_{j'}] = [\tilde{\phi}_{j,j+1}, \tilde{p}_{j',j'+1}] = i\delta_{j,j'}. \tag{146}$$

The Hamiltonian is[40]

$$H_{\mathrm{mV}} = \sum_j \left( \frac{1}{2R^2} p_j^2 + \frac{R^2}{2} \left( \tilde{p}_{j,j+1} + \frac{\phi_{j+1} - \phi_j}{2\pi} \right)^2 \right). \tag{147}$$

We have *electric* Gauss's law constraints at sites and *magnetic* ones on links:

$$G_j \equiv e^{2\pi i g_j} = 1, \qquad \tilde{G}_{j,j+1} \equiv e^{2\pi i \tilde{g}_{j,j+1}} = 1, \tag{148}$$

where $g_j \equiv -p_j + \frac{\tilde{\phi}_{j,j+1} - \tilde{\phi}_{j-1,j}}{2\pi}$ and $\tilde{g}_{j,j+1} \equiv -\tilde{p}_{j,j+1}$, and

$$[g_j, \tilde{g}_{j,j+1}] = [\tilde{g}_{j-1,j}, g_j] = \frac{-i}{2\pi}. \tag{149}$$

**Defects:** The eigenspaces $G_j = e^{i\eta_w}$ and $\tilde{G}_{j,j+1} = e^{i\eta_m}$, respectively, correspond to insertion of winding-symmetry and momentum-symmetry defects. The defects are topological since the unitary operators $e^{-i\eta_m g_j}$ and $e^{-i\eta_w \tilde{g}_{j,j+1}}$ move them by one site. These unitary operators permute different eigenspaces of $G_j$ and $\tilde{G}_{j,j+1}$ while commuting with the Hamiltonian. More precisely, we define a $(\eta_m, \eta_w) \in U(1)_m \times U(1)_w$ defect at link $(j, j+1)$ by modifying the Gauss's law to

$$\tilde{G}_{j,j+1} = e^{i\eta_m}, \ G_{j+1} = e^{i\eta_w}: \quad \mathcal{H}^{(j,j+1)}_{(\eta_m,\eta_w)} = \ \underset{\scriptstyle j-1 \quad j \quad j+1 \quad j+2}{\underbrace{\text{---}\bullet\text{---}\bullet\overset{(\eta_m,\eta_w)}{\times}\bullet\text{---}\bullet\text{---}}}. \tag{150}$$

In this presentation, the insertion of defects corresponds to a modification of the Hilbert space instead of the Hamiltonian, i.e.

$$H^{(j,j+1)}_{(\eta_m,\eta_w)} = H_{\mathrm{mV}}, \qquad \mathcal{H}^{(j,j+1)}_{(\eta_m,\eta_w)} = \left\{ |\psi\rangle : \tilde{G}_{j,j+1}|\psi\rangle = e^{i\eta_m}|\psi\rangle, G_{j+1}|\psi\rangle = e^{i\eta_w}|\psi\rangle \right\}. \tag{151}$$

**Fusion operators:** The fusion operators are given by

$$\lambda^j \left( (\eta_m, \eta_w), (\eta'_m, \eta'_w) \right) = e^{-i[\eta_m] g_j} e^{-i[\eta_w] \tilde{g}_{j,j+1}} \ : \qquad \qquad , \tag{152}$$

[40]The relation with the previous presentation is: $\tilde{\phi}_{j,j+1} = E_{j,j+1}$ and $\tilde{p}_{j,j+1} = -n_{j,j+1}$.

where $0 \leq [\eta] < 2\pi$ such that $[\eta] = \eta \pmod{2\pi}$. Note that the choice of fundamental domain $[0, 2\pi)$ fixes the phase of the fusion operators. For instance, changing $[\eta_m]$ to $[\eta_m] + 2\pi$ multiplies the fusion operator above by $e^{-i2\pi g_j} = e^{-i\eta_w}$.

Let us verify that the fusion operators above indeed implement the fusion correctly. First, note that the fusion operator $\lambda^j\big((\eta_m, \eta_w), (\eta_m', \eta_w')\big)$ commutes with all the Gauss's law constraints, except for $\tilde{G}_{j-1,j}, G_j, \tilde{G}_{j,j+1}, G_{j+1}$. The unitary operator $e^{-i[\eta_w]\tilde{g}_{j,j+1}}$ implements the fusion of the winding-symmetry defects:

$$e^{+i[\eta_w]\tilde{g}_{j,j+1}} G_j \, e^{-i[\eta_w]\tilde{g}_{j,j+1}} = e^{-i\eta_w} G_j, \qquad e^{+i[\eta_w]\tilde{g}_{j,j+1}} G_{j+1} \, e^{-i[\eta_w]\tilde{g}_{j,j+1}} = e^{+i\eta_w} G_{j+1}. \quad (153)$$

Thus, a state $|\psi\rangle$ satisfying $G_j|\psi\rangle = e^{i\eta_w}|\psi\rangle$ and $G_{j+1}|\psi\rangle = e^{i\eta_w'}|\psi\rangle$ gets map to $|\psi'\rangle = \lambda^j\big((\eta_m, \eta_w), (\eta_m', \eta_w')\big)|\psi\rangle$, which satisfies $G_j|\psi'\rangle = |\psi'\rangle$ and $G_{j+1}|\psi'\rangle = e^{i\eta_w + i\eta_w'}|\psi'\rangle$. Similarly for $e^{-i[\eta_m]g_j}$, we have the commutation relations:

$$e^{+i[\eta_m]g_j} \tilde{G}_{j-1,j} \, e^{-i[\eta_m]g_j} = e^{-i\eta_m} \tilde{G}_{j-1,j}, \qquad e^{+i[\eta_m]g_j} \tilde{G}_{j,j+1} \, e^{-i[\eta_m]g_j} = e^{i\eta_m} \tilde{G}_{j,j+1}. \quad (154)$$

**The anomaly:** Using the formula for the anomaly (76), we find

$$G_j^{\frac{-[\eta_m]}{2\pi}} \tilde{G}_{j,j+1}^{\frac{-[\eta_w]}{2\pi}} G_{j-1}^{\frac{-[\eta_m]}{2\pi}} \tilde{G}_{j-1,j}^{\frac{-[\eta_w]}{2\pi}} G_j^{\frac{-[\eta_m']}{2\pi}} \tilde{G}_{j,j+1}^{\frac{-[\eta_w']}{2\pi}} = F(g, g', g'') G_j^{\frac{-[\eta_m+\eta_m']}{2\pi}} \tilde{G}_{j,j+1}^{\frac{-[\eta_w+\eta_w']}{2\pi}} G_{j-1}^{\frac{-[\eta_m]}{2\pi}} \tilde{G}_{j-1,j}^{\frac{-[\eta_w]}{2\pi}}, \quad (155)$$

where we have used the shorthand notations $G_j^{\frac{-[\eta_m]}{2\pi}} \tilde{G}_{j,j+1}^{\frac{-[\eta_w]}{2\pi}} \equiv e^{-i[\eta_m]g_j} e^{-i[\eta_w]\tilde{g}_{j,j+1}}$, $g = (\eta_m, \eta_w)$, $g' = (\eta_m', \eta_w')$, and $g'' = (\eta_m'', \eta_w'')$. Simplifying the equation above, we find

$$F(g, g', g'') = e^{i\{\eta_w, \eta_w'\}(\eta_m + \eta_m')} \tilde{G}_{j,j+1}^{-\{\eta_w, \eta_w'\}} G_j^{-\{\eta_m, \eta_m'\}} : \ \mathcal{H}_{gg'g''}^{(j,j+1)} \to \mathcal{H}_{gg'g''}^{(j,j+1)}, \quad (156)$$

where

$$\{\eta, \eta'\} \equiv \frac{[\eta] + [\eta'] - [\eta + \eta']}{2\pi} = \begin{cases} 0, & \text{for } [\eta] + [\eta'] \leq 2\pi, \\ 1, & \text{for } [\eta] + [\eta'] > 2\pi. \end{cases} \quad (157)$$

In the expression (156), $F(g, g', g'')$ acts on the defect Hilbert space $\mathcal{H}_{gg'g''}^{(j,j+1)}$ where $G_j = 1$ and $\tilde{G}_{j,j+1} = e^{i\eta_m + i\eta_m' + i\eta_m''}$. Therefore, we find the anomaly cocycles

$$\omega(g, g', g'') = \exp\big(-i\{\eta_w, \eta_w'\}\eta_m''\big), \qquad \text{and} \qquad \alpha(g, g') = 1. \quad (158)$$

Since the 2-cocycle $\alpha$ is trivial, there is no LSM anomaly. The 3-cocycle $\omega$ describes a mixed anomaly between $U(1)_m$ and $U(1)_w$ in the sense that the anomaly trivializes by setting $\eta_m = \eta_m' = \eta_m'' = 0$ or $\eta_w = \eta_w' = \eta_w'' = 0$.

By setting $\eta_m = \pm\eta_w$, we find the anomaly in the chiral $U(1)_L$ and $U(1)_R$ subgroups of $U(1)_m \times U(1)_w$ symmetry.[41] In other words, the chiral anomalies are computed as $\omega_L(\eta, \eta', \eta'') = \exp\big(-i\{\eta, \eta'\}\eta''\big)$ and $\omega_R(\eta, \eta', \eta'') = \exp\big(+i\{\eta, \eta'\}\eta''\big)$.

**Charge conjugation symmetry:** Besides $U(1)_m \times U(1)_w$ symmetry, there is also a charge conjugation symmetry $C$ that anti-commutes with the momentum and winding charges. To find the defect, we conjugate the Hamiltonian and Gauss's law operators by the truncated charge conjugation operator $C^{\leq J} = C^J C^{J-1} C^{J-2} \cdots$ acting as

$$\begin{aligned} C^{\leq J} \phi_j (C^{\leq J})^{-1} &= \begin{cases} -\phi_j, & \text{for } j \leq J, \\ \phi_j, & \text{for } j > J, \end{cases} \\[1ex] C^{\leq J} \tilde{\phi}_{j,j+1} (C^{\leq J})^{-1} &= \begin{cases} -\tilde{\phi}_{j,j+1}, & \text{for } j \leq J, \\ \tilde{\phi}_{j,j+1}, & \text{for } j > J. \end{cases} \end{aligned} \quad (159)$$

---

[41]More precisely, we have $\frac{U(1)_L \times U(1)_R}{\mathbb{Z}_2} = U(1)_m \times U(1)_w$.

We now consider topological defects for $(U(1)_m \times U(1)_w) \rtimes \mathbb{Z}_2^C$. We parametrize the symmetry group by $(\eta_m, \eta_w, \epsilon)$ with the "multiplication" rule

$$(\eta_m, \eta_w, \epsilon) \cdot (\eta_m', \eta_w', \epsilon') = (\eta_m + \epsilon \eta_m', \eta_w + \epsilon \eta_w', \epsilon \epsilon'), \tag{160}$$

with the identification $\eta_m \sim \eta_m + 2\pi$ and $\eta_w \sim \eta_w + 2\pi$, and where $\epsilon, \epsilon' = \pm 1$.

The defects are given by

$$H^{(J,J+1)}_{(\eta_m,\eta_w,\epsilon)} = \frac{R^2}{2}\left(\tilde{p}_{J,J+1} + \frac{\epsilon \phi_{J+1} - \phi_J}{2\pi}\right)^2 + \sum_j \frac{1}{2R^2}p_j^2 + \sum_{j \neq J} \frac{R^2}{2}\left(\tilde{p}_{j,j+1} + \frac{\phi_{j+1} - \phi_j}{2\pi}\right)^2,$$

$$\mathcal{H}^{(J,J+1)}_{(\eta_m,\eta_w,\epsilon)} = \left\{ \tilde{G}_{J,J+1} = e^{i\eta_m}, e^{-2\pi i \epsilon p_{J+1} + i\epsilon \tilde{\phi}_{J+1,J+2} - i\tilde{\phi}_{J,J+1}} = e^{i\eta_w}, \tilde{G}_{j,j+1} = G_j = 1 \text{ for } j \neq J \right\}. \tag{161}$$

The fusion operator for these defects are given by

$$\lambda^j((\eta_m, \eta_w, \epsilon), (\eta_m', \eta_w', \epsilon')) = G_j^{\frac{-[\eta_m]}{2\pi}} \tilde{G}_{j,j+1}^{\frac{-[\eta_w]}{2\pi}} (C_j)^\varepsilon, \tag{162}$$

where $\epsilon = (-1)^\varepsilon$, and $C_j = e^{i\frac{\pi}{2}(\phi_j^2 + p_j^2)} e^{i\frac{\pi}{2}(\tilde{\phi}_{j,j+1}^2 + \tilde{p}_{j,j+1}^2)}$ implements a charge conjugation transformation only at site $j$ and link $(j, j+1)$.

The anomaly is given by

$$G_j^{\frac{-[\eta_m]}{2\pi}} \tilde{G}_{j,j+1}^{\frac{-[\eta_w]}{2\pi}} C_j^\varepsilon \, G_{j-1}^{\frac{-[\eta_m]}{2\pi}} \tilde{G}_{j-1,j}^{\frac{-[\eta_w]}{2\pi}} C_{j-1}^\varepsilon \, G_j^{\frac{-[\eta_m']}{2\pi}} \tilde{G}_{j,j+1}^{\frac{-[\eta_w']}{2\pi}} C_j^{\varepsilon'}$$

$$= F(g, g', g'') G_j^{\frac{-[\eta_m + \eta_m']}{2\pi}} \tilde{G}_{j,j+1}^{\frac{-[\eta_w + \eta_w']}{2\pi}} C_j^{\varepsilon + \varepsilon'} \, G_{j-1}^{\frac{-[\eta_m]}{2\pi}} \tilde{G}_{j-1,j}^{\frac{-[\eta_w]}{2\pi}} C_{j-1}^\varepsilon, \tag{163}$$

where $g = (\eta_m, \eta_w, \epsilon)$, $g' = (\eta_m', \eta_w', \epsilon')$, and $g'' = (\eta_m'', \eta_w'', \epsilon'')$. Simplifying the equation above, we find

$$F(g, g', g'') = e^{i\frac{[\eta_w] + \epsilon[\eta_w'] - [\eta_w + \epsilon \eta_w']}{2\pi}(\eta_m + \epsilon \eta_m')} \tilde{G}_{j,j+1}^{-\frac{[\eta_w] + \epsilon[\eta_w'] - [\eta_w + \epsilon \eta_w']}{2\pi}} G_j^{-\frac{[\eta_m] + \epsilon[\eta_m'] - [\eta_m + \epsilon \eta_m']}{2\pi}}. \tag{164}$$

Here, $F(g, g', g'')$ acts on $\mathcal{H}^{(j,j+1)}_{gg'g''}$ where $G_j = 1$ and $\tilde{G}_{j,j+1} = e^{i\eta_m + i\epsilon \eta_m' + i\epsilon \epsilon' \eta_m''}$. Thus

$$\omega(g, g', g'') = \exp\left(-i\frac{[\eta_w] + \epsilon[\eta_w'] - [\eta_w + \epsilon \eta_w']}{2\pi} \epsilon \epsilon' \eta_m''\right), \quad \text{and} \quad \alpha(g, g') = 1. \tag{165}$$

**Chiral gauge theories on the lattice:** We end the discussion by pointing out an application of the gauging procedure described in Section 3.2. Taking multiple copies of the modified Villain theory, there exist chiral $U(1)$ subgroups free of anomalies. Specifically, consider $N$ copies of the modified-Villain model and the conserved charge

$$Q = \sum_{a=1}^{N} \left(m_a Q_m^a + w_a Q_w^a\right). \tag{166}$$

We take $m_a$ and $w_a$ to be integers to ensure that $Q$ generates a $U(1)$ symmetry. The 't Hooft anomaly in this $U(1)$ symmetry is given by

$$\sum_{a=1}^{N} m_a w_a \in \mathbb{Z}. \tag{167}$$

We can choose the integers $m_a$ and $w_a$ such that the anomaly vanishes while $m_a w_a \neq 0$. An example is (the bosonization of) the 34-50 model [123–127], where $Q/2 = 4Q_m^1 - Q_w^1 + 2Q_m^2 + 2Q_w^2$.

In such cases, $Q$ generates a chiral U(1) symmetry that is free of 't Hooft anomalies. By gauging such symmetries, we obtain (the bosonization of) chiral gauge theories on the lattice [128–132].[42] We leave the details and explicit Hamiltonian of such theories for a future work.

## 4.4  Fermionic theories – N Majorana fermions

Here we discuss anomalies of fermionic theories. In such theories, there is a $\mathbb{Z}_2^{\mathrm{F}}$ fermion-parity symmetry, or more precisely, a $\mathbb{Z}_2^{\mathrm{F}}$ automorphism of the operator algebra. Usually, this automorphism is an inner automorphism and is denoted by $(-1)^{\mathrm{F}}$. The notion of locality in fermionic theories differs from bosonic ones since two fermionic operators with disjoint support anti-commute with each other. Below, we consider systems with a lattice translation symmetry that we denote by $\mathbb{Z}_{\mathrm{T}}$, and an internal symmetry group $G$. We denote the bosonic "part" of the symmetry by $G_{\mathrm{b}} = G/\mathbb{Z}_2^{\mathrm{F}}$.

Following the ideas from [59] and [68], we claim that there are three 'layers' of anomalies in 1+1d fermionic systems:

1. The first layer is classified by $H^1(G_{\mathrm{b}} \times \mathbb{Z}_{\mathrm{T}}, \mathbb{Z}_2^{\mathrm{Arf}}) = H^1(G_{\mathrm{b}}, \mathbb{Z}_2^{\mathrm{Arf}}) \times H^1(\mathbb{Z}_{\mathrm{T}}, \mathbb{Z}_2^{\mathrm{Arf}})$ and is measured by the number of Majorana zero modes modulo 2 localized on the defects. It is sometimes referred to as the Arf layer since it corresponds to the 0+1d anomaly of $(-1)^{\mathrm{F}}$ whose bulk 1+1d SPT realization is given by the Arf invariant of 2-dimensional spin manifolds.

2. The second later is classified by $H^2(G_{\mathrm{b}} \times \mathbb{Z}_{\mathrm{T}}, \mathbb{Z}_2^{\mathrm{F}})$. This layer captures the mixed-anomaly between $G_{\mathrm{b}} \times \mathbb{Z}_{\mathrm{T}}$ and $\mathbb{Z}_2^{\mathrm{F}}$, which has a term valued in $H^2(G_{\mathrm{b}}, \mathbb{Z}_2^{\mathrm{F}})$ capturing the mixed anomaly between $G_{\mathrm{b}}$ and $\mathbb{Z}_2^{\mathrm{F}}$, and another term valued in $H^1(G_{\mathrm{b}}, \mathbb{Z}_2)$ capturing the cubic mixed-anomaly between all three symmetries. The latter is a fermionic generalization of the LSM anomaly. This anomaly is given by the fermion parity of the fusion operators. Let us define the fermion parity of the fusion operators by $K(g_1, g_2) = \pm 1$, where

$$(-1)^{\mathrm{F}} \lambda^j(g_1, g_2)(-1)^{\mathrm{F}} = K(g_1, g_2)\lambda^j(g_1, g_2). \tag{168}$$

Acting with $(-1)^{\mathrm{F}}$ on equation (76), we find that $K$ satisfies a modified cocycle condition

$$\frac{K(g_2, g_3)K(g_1, g_2 g_3)}{K(g_1 g_2, g_3)K(g_1, g_2)} = K(g_1, 1). \tag{169}$$

Moreover, by conjugating the defect Hamiltonian $H_g^{(j,j+1)}$ by a local unitary operator of fermion parity $\gamma(g) = \pm 1$, we find the identification[43]

$$K(g_1, g_2) \sim K(g_1, g_2)\frac{\gamma(g_1)\gamma(g_2)}{\gamma(g_1 g_2)}. \tag{171}$$

These leads to standard cocycle conditions for

$$\beta(g_1, g_2) \equiv \frac{K(g_1, g_2)}{K(g_1, 1)}, \qquad \text{and} \qquad f(g) \equiv K(g, 1). \tag{172}$$

---

[42]We thank Edward Witten and Nati Seiberg for discussions on this point.

[43]Without a translation symmetry we have

$$\frac{K^j(g_2, g_3)K^j(g_1, g_2 g_3)}{K^j(g_1 g_2, g_3)K^{j-1}(g_1, g_2)} = K^{j-1}(g_1, 1), \quad \text{and} \quad K^j(g_1, g_2) \sim K^j(g_1, g_2)\frac{\gamma^{j-1}(g_1)\gamma^j(g_2)}{\gamma^j(g_1 g_2)}. \tag{170}$$

Here, $f : G \rightarrow \mathbb{Z}_2$ is a group homomorphism that measures whether the movement operator for a $g$-defect is bosonic or fermionic. We note that by conjugating defect Hamiltonian by fermionic local operators the 2-cocycle $\beta$ is shifted by an exact term. To get the anomaly we mod out the cocycles by the freedom of redefining the defect Hamiltonians by local unitaries and find the cohomology classes

$$[\beta] \in H^2(G_\text{b}, \mathbb{Z}_2^\text{F}), \qquad \text{and} \qquad f \in H^1(G_\text{b}, \mathbb{Z}_2^\text{F}). \tag{173}$$

Here, $[\beta]$ is a mixed anomaly between $G_\text{b}$ and $\mathbb{Z}_2^\text{F}$. The homomorphism $f$ is a fermionic generalization of LSM anomaly that involves translation, $G_\text{b}$, and $\mathbb{Z}_2^\text{F}$.

3. The third layer is just the bosonic layer that has been discussed in length before. It is classified by $H^3(G_\text{b} \times \mathbb{Z}_\text{T}, U(1)) = H^3(G_\text{b}, U(1)) \times H^2(G_\text{b}, U(1))$.

Putting everything together the anomaly in $G_\text{b} \times \mathbb{Z}_\text{T} \times \mathbb{Z}_2^\text{F}$ is classified by

$$H^1(G_\text{b}, \mathbb{Z}_2^\text{Arf}) \times H^1(\mathbb{Z}_\text{T}, \mathbb{Z}_2^\text{Arf}) \times H^2(G_\text{b}, \mathbb{Z}_2^\text{F}) \times H^1(G_\text{b}, \mathbb{Z}_2^\text{F}) \times H^3(G_\text{b}, U(1)) \times H^2(G_\text{b}, U(1)). \tag{174}$$

Forgetting the translation symmetry, this classification is compatible with [133]. See [45], for a discussion of fermionic LSM anomalies for on-site internal symmetries. As a concrete example, below we consider the chain of $N$ free Majorana fermions.

**Free Majorana fermions**

Consider a chain of $N$ flavor of free Majorana fermions with the Hamiltonian

$$H = i \sum_j \sum_{a=1}^N \psi_j^a \psi_{j+1}^a, \tag{175}$$

where $\{\psi_j^a, \psi_{j'}^b\} = 2\delta_{j,j'}\delta_{a,b}$, and $a$ is a flavor index. The model has a $\mathbb{Z}_\text{T}$ lattice translation symmetry $T$ acting as

$$T : \psi_j^a \mapsto \psi_{j+1}^a. \tag{176}$$

There is also a global $O(N)$ symmetry that rotates $\psi_j^a$ at a given site among each other, i.e.

$$U_R : \psi_j^a \mapsto \sum_{b=1}^N R_{ab} \psi_j^b, \tag{177}$$

for an $N \times N$ orthonormal matrix $R$. Here, $\mathbb{Z}_2^\text{F} \in O(N)$ is generated by the central element $(-1)^\text{F} = U_{-\mathbb{1}}$ that acts as $\psi_j^a \mapsto -\psi_j^a$. The bosonic symmetry group is given by the quotient $G_\text{b} = O(N)/\mathbb{Z}_2^\text{F}$.

We note that every element of $O(N)$ is a product of at most $N$ "reflections". Therefore, we can generate the $O(N)$ global symmetry by reflections. Reflection along the plane perpendicular to a vector $\vec{v} \in \mathbb{R}^N$, up to an overall sign, is generated by the unitary operator

$$U_{\vec{v}} = \prod_j \left( \sum_{a=1}^N v_a \psi_j^a \right), \quad \text{where} \quad \sum_{a=1}^N v_a^2 = 1. \tag{178}$$

More precisely,

$$U_{\vec{v}} (x_a \psi_j^a) U_{\vec{v}}^{-1} = (-1)^L \left( x_a \psi_j^a - 2(\vec{x} \cdot \vec{v}) v_a \psi_j^a \right), \tag{179}$$

where $L$ is the number of sites. Therefore, up to the action of $(-1)^\text{F}$, the global $O(N)$ symmetry is generated by $U_{\vec{v}}$. In another words, the quotient symmetry $G_b = O(N)/\mathbb{Z}_2^\text{F}$ is generated by

$$U_R = U_{\vec{v}^1} U_{\vec{v}^2} \cdots U_{\vec{v}^n}, \tag{180}$$

for $n \leq N$. Here, $R \in O(N)$ correspond to an $N \times N$ matrix given by the product of $n$ reflections along the planes perpendicular to the vectors $\vec{v}^k$ for $k = 1, 2, \cdots, n$. This system also has reflection and time-reversal symmetries that we do not discuss here; see [43, 46] for a discussion.[44]

We can construct the defects by conjugating the system with the truncated symmetry operator. However, since the symmetry is a product of "local" factors, the fusion operators take the simple form

$$\lambda^j(\vec{v}^1 \vec{v}^2 \cdots \vec{v}^n, \vec{w}^1 \vec{w}^2 \cdots \vec{w}^m) = \prod_{k=1}^{n} \left( \sum_{a=1}^{N} v_a^k \psi_j^a \right), \tag{181}$$

where $v_a^k = (\vec{v}^k)_a$.

**Anomalies**

Now let us find the anomalies involving the $O(N) \times \mathbb{Z}_{\mathrm{T}}$ symmetry. Specifically, we discuss the three layers of anomaly involving $O(N)/\mathbb{Z}_2^{\mathrm{F}} \times \mathbb{Z}_{\mathrm{T}}$.

**First layer:** The first layer corresponds to the number of Majorana zero modes on the symmetry defects. For the internal symmetry $G_{\mathrm{b}} = O(N)/\mathbb{Z}_2^{\mathrm{F}}$, the defect Hilbert space is the same as the original Hilbert space since the symmetry is written as a product of local terms. Therefore, there are no extra degrees of freedom and no Majorana zero modes on the internal symmetry defects. Therefore, there is no $H^1(G_{\mathrm{b}}, \mathbb{Z}_2^{\mathrm{Arf}})$ anomaly.

The other anomaly is classified by $H^1(\mathbb{Z}_{\mathrm{T}}, \mathbb{Z}_2^{\mathrm{Arf}}) \cong \mathbb{Z}_2$. This counts the number of Majorana zero modes per site which is equal to the number of flavors $N \pmod 2$. To see this, we note that at each site we have a Clifford algebra of rank $N$ which has a center for odd $N$. When $N$ is odd, the central element, $\psi_j^1 \psi_j^2 \cdots \psi_j^N$, is the Majorana zero mode at site $j$. Therefore, there is an anomaly involving fermion parity $\mathbb{Z}_2^{\mathrm{F}}$ and lattice translation when $N$ is odd.

**Second layer:** The anomaly in the second layer corresponds to the fermion parity of the fusion operators. We find that when $n$ is odd and the internal symmetry is a product of an odd number of reflections (i.e. $\det(R) = -1$), then the fusion operator is fermionic and otherwise it is bosonic – see equation (181). This means that $K(R_1, R_2) = \det(R_1)$ in equation (168). Therefore, $\beta = 1$ and $f(R) = \det(R)$. Thus, we conclude that there is no anomaly involving $O(N)/\mathbb{Z}_2^{\mathrm{F}}$ and $\mathbb{Z}_2^{\mathrm{F}}$. However, there is a cubic mixed anomaly involving $O(N)/\mathbb{Z}_2^{\mathrm{F}}$, $\mathbb{Z}_2^{\mathrm{F}}$, and lattice translation which is given by $\det(R)$. This anomaly gets trivialized if we consider the $SO(N)$ subgroup of $O(N)$.

**Third layer:** The third layer is the bosonic layer of the anomaly we discussed earlier. Since the internal symmetry acts "on-site", this anomaly is captured by the projectivity of the action of the symmetry on each site. We note that the Clifford algebra at site $j$ generates the group $\mathrm{Pin}^+(N)$ which is a double cover of $O(N)$. Specifically, the faithful symmetry group is $O(N) = \mathrm{Pin}^+(N)/\{\pm 1\}$. When $N \geq 3$, $\mathrm{Pin}^+(N)$ is a non-trivial $\mathbb{Z}_2$ extension of $O(N)$ and therefore the symmetry is realized projectively per site. This leads to an LSM-type anomaly for the $O(N) \times \mathbb{Z}_{\mathrm{T}}$ symmetry when $N \geq 3$.

We note that our anomaly detection method is local; hence, we do not specify the chain details, such as the number of sites and the boundary conditions. Relatedly, we do not specify the Hilbert space of the theory and instead focus on the algebra of local operators. However, for a finite chain, we construct the Hilbert space as an irreducible representation of the operator algebra. In some cases, the operator algebra has a center which leads to different irreducible

---

[44]See also [134] for a discussion of the lattice translation symmetry in the Schwinger model.

representations. The eigenvalue of a central element should be viewed as a *theta* parameter. Alternatively, one can sum over such theta parameters (which corresponds to gauging a $(-1)$-form symmetry). For instance, in a system with an odd number of Majorana fermions, there is a central element given by the product of all Majorana operators. This central element leads to a $\mathbb{Z}_2$ valued theta parameter which is odd under the fermion parity automorphism $(-1)^{\mathrm{F}}$. Thus, the theta parameter breaks $\mathbb{Z}_2^{\mathrm{F}}$. Alternatively, we can consider a larger Hilbert space by summing over the two values of the theta parameter and restoring the $\mathbb{Z}_2^{\mathrm{F}}$ global symmetry. For a detailed discussion of such issues see [43, 46].

**Gauging and bosonization**

As discussed above, the $O(N)$ internal symmetry has no pure anomaly and can be gauged. However, there can be a mixed anomaly involving $O(N)$ and lattice translation. Therefore, the original lattice translation symmetry is violated upon gauging anomalous subgroups of $O(N)$. However, the lattice translation symmetry in such cases appears as a non-invertible symmetry in the theory after gauging. For instance, consider the $\mathbb{Z}_2$ symmetry generated by a single $O(N)$ reflection. Gauging such a symmetry must violate the original translation symmetry.

Here, we consider the case of $N = 1$ and gauge the $(-1)^{\mathrm{F}}$ symmetry following our gauging procedure. We add $\mathbb{Z}_2$ gauge variables on links represented by Pauli matrices $\sigma_{j,j+1}^x, \sigma_{j,j+1}^y, \sigma_{j,j+1}^z$, and find the extended Hamiltonian

$$\tilde{H} = i \sum_j \psi_j \sigma_{j,j+1}^x \psi_{j+1}\,. \tag{182}$$

Using the fusion operators (181), we find the Gauss's law constraints

$$\sigma_{j-1,j}^z \psi_j \sigma_{j,j+1}^z = 1\,. \tag{183}$$

However, due to the anomaly, these Gauss's law constraints do not commute with each other at different sites. However, we modify Gauss's law constraints such that they commute with each other at the expense of violating the translation symmetry. To do such, we first need to make the fusion operators bosonic.

The trick is to add a Majorana fermion $\chi_j$ at each site, and then project them out by considering the constraints

$$i\chi_{2l}\chi_{2l+1} = 1\,. \tag{184}$$

We assume the anti-commutation relation $\{\psi_j, \chi_{j'}\} = 0$ and $\{\chi_j, \chi_{j'}\} = 2\delta_{j,j'}$. The Majorana fermions $\chi_j$ have trivial bulk dynamics and define an invertible fermionic theory known as the Kitaev chain [135]. This invertible theory serves as a choice of discrete torsion for gauging $\mathbb{Z}_2^{\mathrm{F}}$.

Using the added Majorana fermions, we find the modified Gauss's law constraints[45]

$$\mathcal{G}_j \equiv i\sigma_{j-1,j}^z \psi_j \chi_j \sigma_{j,j+1}^z = 1\,, \qquad \text{and} \qquad i\chi_{2l}\chi_{2l+1} = \sigma_{2l,2l+1}^x\,. \tag{185}$$

These local constraints commute with each other. However, they break lattice translation symmetry by one site and only preserve translations by an even number of sites. Below, we do gauge fixing and simplify the gauged theory.

By performing gauge transformations, we rotate the Majorana fermions such that

$$i\psi_{2l-1}\psi_{2l} = -1\,. \tag{186}$$

Using this choice of gauge, we find the Hamiltonian of the theory after gauging to be

$$\tilde{H} = -\sum_l \left( \sigma_{2l-1,2l}^x + \sigma_{2l-1,2l}^z \sigma_{2l+1,2l+2}^z \right)\,. \tag{187}$$

---

[45]The auxiliary Majorana fermions $\chi_j$ allow us to gauge any subgroup of the $O(N)$ flavor symmetry.

We identify this as the familiar transverse-field Ising model on a chain with half the number of sites compared to the Majorana chain. We have done the microscopic version of the bosonization on the lattice. This prescription is more general than the Jordan-Wigner transformation and can be applied to any fermionic lattice Hamiltonian system with an anomaly free $\mathbb{Z}_2$ internal symmetry.

Finally, the translation symmetry operator $T$ of the Majorana chain does not commute with the Gauss's law constraints (185) and therefore is not gauge invariant. By taking its gauge-invariant matrix elements, we find a non-invertible lattice translation symmetry.[46] For a detailed discussion of this non-invertible symmetry see [46, 137].

## Acknowledgments

I am grateful to M. Cheng, A. Cherman, I. Cirac, T. Dumitrescu, D. V. Else, P. Gorantla, Z. Komargodski, M. Levin, Y.-M. Lu, A. P. Polychronakos, A. Prem, T. Senthil, S.-H. Shao, N. Sopenko, Y. Tachikawa, and E. Witten for discussions. I especially thank Nati Seiberg and Wilbur Shirley for numerous illuminating discussions. I thank Michael Levin, Nati Seiberg, and Yuji Tachikawa for their comments on the draft.

**Funding information** The author gratefully acknowledges support from U.S. Department of Energy grant DE-SC0009988.

## A Anomaly in $\mathbb{Z}_2$ – Frobenius-Schur indicator

In this appendix, we provide a simple anomaly indicator for internal $\mathbb{Z}_2$ symmetries. Consider a topological defect $a$ with a $\mathbb{Z}_2$ fusion rules, i.e. $a^{-1} = a$. Here we can find its anomaly with a single equation. Specifically, we compute the Frobenius-Schur (FS) indicator of the topological defect on the lattice (see e.g. [138] for the definition of FS indicator).

The anomaly of a $\mathbb{Z}_2$ symmetry $a$ is given by an invariant $\chi_a = \pm 1$ (i.e. its Frobenius-Schur indicator), which is given by[47]

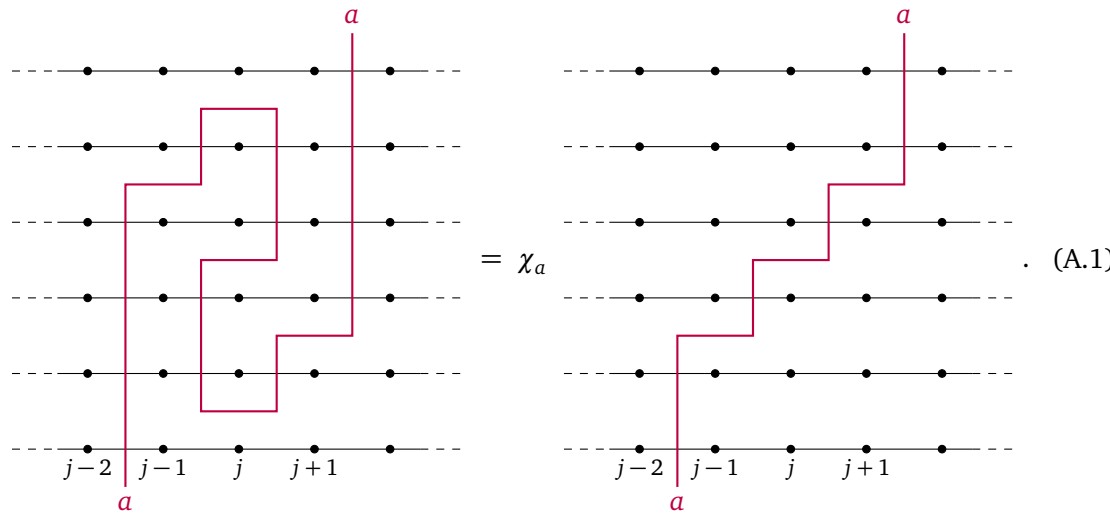

$$= \chi_a \qquad . \tag{A.1}$$

---

[46]For the continuum analog of this non-invertible symmetry see [136].
[47]This relation is the lattice analog of equation (187) of [138].

In equation, we have

$$\lambda^j(a,a)\lambda^{j-1}(a,1)\lambda^j(a,1)\lambda^{j+1}(a,1)\big(\lambda^j(a,a)\big)^{-1} = \chi_a\,\lambda^{j+1}(a,1)\lambda^j(a,1)\lambda^{j-1}(a,1). \quad \text{(A.2)}$$

From this equation, we see that $\chi_a$ is "gauge invariant" in the sense that it does not change under phase redefinition of the fusion operators. Therefore $\chi_a$ is identified with the anomaly of the $\mathbb{Z}_2$ symmetry inside $H^3(\mathbb{Z}_2, U(1)) \cong \mathbb{Z}_2$.

For our choice the anomaly 3-cocycle we have $\chi_a = \omega(a,a,a)$, which can be seen using the fact that $\lambda^j$ commute with $\lambda^{j+2}$. Note that in our construction, the anomaly 3-cocycle satisfies the relation $\omega(1,g_2,g_3) = \omega(g_1,1,g_3) = \omega(g_1,g_2,1) = 1$. Hence, for a $\mathbb{Z}_2$ symmetry $\omega(a,a,a)$ captures the anomaly since it is the only non-trivial element of the anomaly 3-cocycle.

## B  Comparison with Else-Nayak [1]

In this appendix, we show when the anomaly detection method of Else and Nayak [1] works, our method gives the same result as theirs.[48] The Else-Nayak method works for internal symmetries that are realized by finite-depth local unitaries. Our method can capture the anomaly in any locality-preserving symmetry that is not necessarily realized by a finite-depth quantum circuit. Moreover, we can also detect *mixed* anomalies involving internal symmetries and any other symmetry, such as reflection that is not even locality-preserving.

Here, we consider an internal symmetry $G$, where each symmetry operator is written as a product of local terms. Specifically, we have

$$U_g = \cdots U_g^{j+2} U_g^{j+1} U_g^j U_g^{j-1} U_g^{j-2} \cdots, \quad \text{(B.1)}$$

where $U_g^j$ is a local operator with finite support around site $j$. For simplicity, we assume that $U_g^j$ has support on at most sites $j$ and $j+1$. However, the conclusion of this section holds for the more general case where $U_g^j$ has support at most on sites $j, j+1, \cdots, j+k$, for a fixed integer $k$.

The local factors $U_g^j$ above are the movement operators of symmetry defects. In terms of the truncated symmetry operators, they are given by $U_g^j = U_g^{\leq j}\big(U_g^{\leq j-1}\big)^{-1}$; see equation (47). The defect Hamiltonian for two defects is given by

$$H_{g_1;g_2}^{(j-1,j);(j,j+1)} = U_{g_1}^{\leq j-1} U_{g_2}^{\leq j} H \big(U_{g_2}^{\leq j}\big)^{-1} \big(U_{g_1}^{\leq j-1}\big)^{-1}. \quad \text{(B.2)}$$

Therefore, the fusion operators are given by

$$\lambda^j(g_1,g_2) = U_{g_1 g_2}^{\leq j}\big(U_{g_2}^{\leq j}\big)^{-1}\big(U_{g_1}^{\leq j-1}\big)^{-1}. \quad \text{(B.3)}$$

In [1], the anomaly cocycle is found by considering restricted symmetry operators that act on a finite region in space. In other words, they consider the truncated operators

$$U_g^{[J_1,J_2]} = U_g^{\leq J_2}\big(U_g^{\leq J_1}\big)^{-1}, \quad \text{(B.4)}$$

that act on the interval $[J_1, J_2]$, where $J_1 < J_2$. Such a restricted symmetry operator represents a $G$ action up to corrections localized around site $J_1$ and $J_2$

$$U_{g_1}^{[J_1,J_2]} U_{g_2}^{[J_1,J_2]} = \big(\Omega_{\mathrm{L}}^{J_1}(g_1,g_2)\Omega_{\mathrm{R}}^{J_2}(g_1,g_2)\big)^{-1} U_{g_1 g_2}^{[J_1,J_2]}. \quad \text{(B.5)}$$

---

[48]We thank Wilbur Shirley for motivating us to write this appendix.

Here, $\Omega_{\text{L}}^{J}(g_1, g_2)$ and $\Omega_{\text{R}}^{J}(g_1, g_2)$ are unitary operators that have support around site $J$. Note that $\left(\Omega_{\text{L}}^{J_1}(g_1, g_2)\Omega_{\text{R}}^{J_2}(g_1, g_2)\right)^{-1}$ here is the same as $\Omega_M(g_1, g_2)$ in [1].

We can write $\Omega_{\text{R}}^{j}$ in terms of the truncated symmetry operators as

$$\Omega_{\text{R}}^{j}(g_1, g_2) = U_{g_1 g_2}^{\leq j}\left(U_{g_2}^{\leq j}\right)^{-1}\left(U_{g_1}^{\leq j}\right)^{-1}. \tag{B.6}$$

Comparing this equation with (B.3), we find

$$\Omega_{\text{R}}^{j}(g_1, g_2) = \lambda^j(g_1, g_2)\left(\lambda^j(g_1, 1)\right)^{-1}. \tag{B.7}$$

Now we will find a formula for the anomaly in terms of $\Omega_{\text{R}}^{j}$.

Using equation (76), we find

$$\omega^j(g_1, g_2, g_3) = \frac{\lambda^j(g_1, g_2 g_3)\lambda^{j-1}(g_1, 1)\lambda^j(g_2, g_3)\left(\lambda^j(g_2, 1)\right)^{-1}}{\lambda^j(g_1 g_2, g_3)\left(\lambda^j(g_1 g_2, 1)\right)^{-1}\lambda^j(g_1, g_2)\lambda^{j-1}(g_1, 1)}, \tag{B.8}$$

which in terms of $\Omega_{\text{R}}^{j}$ can be written as

$$\omega^j(g_1, g_2, g_3) = \frac{\Omega_{\text{R}}^{j}(g_1, g_2 g_3)\lambda^j(g_1, 1)\lambda^{j-1}(g_1, 1)\Omega_{\text{R}}^{j}(g_2, g_3)}{\Omega_{\text{R}}^{j}(g_1 g_2, g_3)\Omega_{\text{R}}^{j}(g_1, g_2)\lambda^j(g_1, 1)\lambda^{j-1}(g_1, 1)}. \tag{B.9}$$

Since $U_g^j = \lambda^j(g, 1)$ has support on at most sites $j$ and $j+1$

$$U_{g_1}^{\leq j}\Omega_{\text{R}}^{j}(g_2, g_3)\left(U_{g_1}^{\leq j}\right)^{-1} = \lambda^j(g_1, 1)\lambda^{j-1}(g_1, 1)\Omega_{\text{R}}^{j}(g_2, g_3)\left(\lambda^j(g_1, 1)\lambda^{j-1}(g_1, 1)\right)^{-1}, \tag{B.10}$$

thus we find

$$\Omega_{\text{R}}^{j}(g_1 g_2, g_3)\Omega_{\text{R}}^{j}(g_1, g_2) = \omega^j(g_1, g_2, g_3)\Omega_{\text{R}}^{j}(g_1, g_2 g_3)U_{g_1}^{\leq j}\Omega_{\text{R}}^{j}(g_2, g_3)\left(U_{g_1}^{\leq j}\right)^{-1}. \tag{B.11}$$

The equation above is the same as equation (5) of [1]. Therefore, our 3-cocycle $\omega^j$ coincides with the one computed by Else and Nayak. Equation (B.11) for the anomaly 3-cocycle has also appeared in the context of AQFT; see, for instance, [80, Section 4.2].

## C   Defects for antiunitary symmetries

As shown in Section 2.4, any locality-preserving symmetry operator is representable by a topological defect. In particular, we showed that defects for lattice translation correspond to adding or removing a site. Here, we construct topological defects for antiunitary symmetries.

Consider a system where the Hilbert space is a tensor product of local Hilbert spaces

$$\mathcal{H} = \mathcal{H}_1 \otimes_{\mathbb{C}} \mathcal{H}_2 \otimes_{\mathbb{C}} \cdots \otimes_{\mathbb{C}} \mathcal{H}_L. \tag{C.1}$$

We have used the subscript $\mathbb{C}$ in $\otimes_{\mathbb{C}}$ to emphasize that the tensor product is taken over the complex numbers. In other words, phase factors (or more generally $\mathbb{C}$-numbers) acting on $\mathcal{H}_j$ are identified with those acting on $\mathcal{H}_{j'}$.

Equivalently, we can consider the tensor product over real numbers and identify the phases at different sites by additional local constraints. Namely, we take the real vector space[49]

$$\mathcal{H}^{\mathbb{R}} \equiv \mathcal{H}_1 \otimes_{\mathbb{R}} \mathcal{H}_2 \otimes_{\mathbb{R}} \cdots \otimes_{\mathbb{R}} \mathcal{H}_L, \tag{C.2}$$

---

[49]For tensor product over real numbers, we identify the operator $rO_j \otimes_{\mathbb{R}} O'_j$ with $O \otimes_{\mathbb{R}} rO'$ only for real numbers $r \in \mathbb{R}$, and not for an imaginary number $r$.

and impose the local constraints $u_j(\theta) = u_{j+1}(\theta)$ to get back $\mathcal{H}$. Here, $u_j(\theta)$ implements a $U(1)$ "phase" rotation at site $j$. In other words, $u_j(\theta)|\psi\rangle_{j'} = e^{i\delta_{j,j'}\theta}|\psi\rangle_{j'}$.[50]

Consider the antiunitary symmetry operator $U = \mathcal{K}$, where $\mathcal{K}$ is complex conjugation in a fixed basis. A defect for this symmetry at link $(j, j+1)$ is given by changing the constraint $u_j(\theta) = u_{j+1}(\theta)$ to $u_j(\theta) = u_{j+1}(\theta)^{-1}$. The movement operator $U_j = \mathcal{K}_j$ that moves the defect from link $(j-1, j)$ to $(j, j+1)$ acts as

$$\mathcal{K}_j u_j(\theta)(\mathcal{K}_j)^{-1} = u_j(-\theta), \tag{C.3}$$

and $\mathcal{K}_j u_{j'}(\theta)(\mathcal{K}_j)^{-1} = u_{j'}(\theta)$ for $j' \neq j$. Thus, $\mathcal{K}_j$ acts antiunitarily only on site $j$. We now explain how this is derived from our general construction in Section 2.4.

To find the defect for general antiunitary symmetries, we want to extend the unitary operators to $U_A U_B^{-1}$, where $U_A$ acts only on system $A$ and $U_B$ only acts on system $B$. In other words, we want $U_A$ to act antiunitarily only on system $A$. To do this, we consider the extended Hilbert space and Hamiltonian

$$\tilde{\mathcal{H}} = \mathcal{H}^{\mathbb{R}} \otimes_{\mathbb{R}} \mathcal{H}^{\mathbb{R}}, \qquad \text{and} \qquad \tilde{H} = H \otimes_{\mathbb{R}} \mathbb{1}, \tag{C.4}$$

and the extended symmetry operator

$$\tilde{U} = U \otimes_{\mathbb{R}} U^{-1}. \tag{C.5}$$

The local constraints for the untwisted system are

$$\mathbb{1} \otimes_{\mathbb{R}} \pi_j = 1, \quad u_j(\theta) \otimes_{\mathbb{R}} u_j(\theta)^{-1} = 1, \quad \text{and} \quad u_j(\theta)u_{j+1}(\theta)^{-1} \otimes_{\mathbb{R}} \mathbb{1} = 1. \tag{C.6}$$

The first constraint just projects out system $B$ to one of its states. The second and third constraints reproduce the tensor product over complex numbers.

The system with a defect at link $(J, J+1)$ is described by the defect Hamiltonian (59), and the defect Hilbert space is given by imposing the constraints $\tilde{U}^{\leq J}(\mathbb{1} \otimes_{\mathbb{R}} \pi_j)(\tilde{U}^{\leq J})^{-1} = 1$,

$$\tilde{U}^{\leq J}\left(u_j(\theta) \otimes_{\mathbb{R}} u_j(\theta)^{-1}\right)(\tilde{U}^{\leq J})^{-1} = 1, \quad \text{and} \quad \tilde{U}^{\leq J}\left(u_j(\theta)u_{j+1}(\theta)^{-1} \otimes_{\mathbb{R}} \mathbb{1}\right)(\tilde{U}^{\leq J})^{-1} = 1. \tag{C.7}$$

Now let us apply this to the case of an antiunitary symmetry acting on-site, that is

$$U = \mathcal{K}\prod_j U_j, \tag{C.8}$$

where $U_j$ acts unitarily on site $j$. Since $U_j$ is a complex operator, it commutes with $u_j(\theta)$. Thus, we find the local constraints

$$u_j(\theta) \otimes_{\mathbb{R}} u_j(\theta)^{-1} = 1, \quad \text{and} \quad u_j(\theta)u_{j+1}(\theta)^{-(-1)^{\delta_{j,J}}} \otimes_{\mathbb{R}} \mathbb{1} = 1, \tag{C.9}$$

among other constraints. Hence, the movement operator $U_J$ act antiunitarily only at site $J$. Using this prescription, we can compute the anomaly of antiunitary symmetries and identify the anomaly as the obstruction to gauging. However, gauging an antiunitary symmetry leads to a theory whose Hilbert space is a real vector space!

---

[50]The dimension of $\mathcal{H}^{\mathbb{R}}$ is $2^{L-1}$ times larger than that of $\mathcal{H}$ and there exist $L-1$ independent constraints.

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
