# Peer review of "Lieb-Schultz-Mattis anomalies as obstructions to gauging (non-on-site) symmetries"

_SciPost Physics, doi:SciPost Phys. 16, 098 (2024)_

## Round 2 · Referee Report · Anonymous (Referee 1) · 2023-12-25

Strengths

  1. The authors develop a solid theory of LSM anomalies from the perspective of gauging. They clarify some subtleties left in the previous literature. Discussions on symmetry operators, topological defects, and their relations are quite clear, and easy to follow. 2.The systematic constructions for truncating a non-onsite symmetry, e.g. translations, provide stable foundations for some intuitive ideas. 3.The gauging procedure provides perspectives for other areas, such as lattice gauge theory and non-invertible symmetry.

Report

The paper presents a systematic study of anomaly for both onsite and non-onsite symmetries, from the perspective of gauging obstructions. Some mathematical formalisms are developed to make this gauging procedure clear. Cocycle data can be extracted, providing direct evidence for ungaugability.
The systematic constructions for gauging a non-site symmetry make this appliable to many known models.
I would recommend acceptance.

Requested changes

1. The F-symbol satisfies a modified pentagon equation, right hand side of Eqn. (3.5). Is this specialized to this context or universal on the lattice? Maybe a footnote for clarification is needed.
2. In Sec. 4.1 and 4.2, examples for gauging internal symmetry are given. A short parallel discussion on gauging translation symmetry would confirm the mixed ‘t Hooft anomaly between them.
3. In the third paragraph of page 41, I think there is a logical gap for people who are not familiar with splitable states. “The fact that each region has d > 1 ground states is in contradiction with …” This sentence should be sharpened to make this contradiction clear.

---

## Round 2 · Referee Report · Anonymous (Referee 2) · 2024-1-30

Strengths

  1. A very detailed discussion on various properties of the topological defects for internal and translation symmetry on the lattice such as moving and fusing of the defects.
  2. A concrete procedure to construct topological defects on the lattice.
  3. Discussion on how to obtain the symmetry operators from the defects and vice versa.
  4. Provide a concrete formula to compute the anomalies for internal and spatial symmetries on the lattice.
  5. A very clear discussion on the general gauging procedure and on the anomaly as obstruction of gauging.

Report

This paper provides a very detailed discussion on the topological defects for both internal and translation symmetry and how to compute the anomalies on the lattice. In addition to the strengths mentioned above, it contains a very nice discussion on the topological defects for the lattice translation symmetry and antiunitary symmetry. I believe the framework developed in this work makes an important contribution to our understanding of the topological defect on the lattice. This paper is very well-written and easy to follow. I recommend this paper to be published in SciPost Physics.

Requested changes

  1. The author claims that the method introduced in this work can also be applied to reflection symmetry but doesn't provide explicit discussion. It would be nice to have an example for the reflection symmetry defect.
  2. What modifications for the lattice calculation of the anomaly are needed if the anomaly involves reflection symmetry.
  3. It would be nice to have an example on gauging translation symmetry explicitly on the lattice.

---

## Round 3 · Author Response

We thank both referees for their detailed and helpful comments on our manuscript. We have addressed them in the new version of the draft. See below for a summary of the changes.

---

## Round 3 · List of Changes

Report 1:

  1. Footnote 29 has been added.
  2. Footnote 31 has been added.
  3. We have edited Section 3.3 and added various footnotes to address the referee's point.

Report 2:

  1. We only argue that we can detect the mixed anomaly between an internal symmetry and the reflection symmetry. Since the reflection symmetry is not locality-preserving (in the sense explained in Section 2), we do not know how to construct a topological defect for it.
  2. A paragraph at the end of Subsection 3.1 has been added, commenting on the anomaly involving reflection.
  3. We do not know how to gauge a symmetry that does not act internally, such as lattice translation. See also the new footnote 31.

---

## Editorial Decision

published